# ContScout: sensitive detection and removal of contamination from annotated genomes

**Balázs Bálint** [1], **Zsolt Merényi** [1], **Botond Hegedüs** [1], **Igor V. Grigoriev** [2,3], **Zhihao Hou** [1,4], **Csenge Földi** [1,4] & **László G. Nagy** [1] ✉

Contamination of genomes is an increasingly recognized problem affecting several downstream applications, from comparative evolutionary genomics to metagenomics. Here we introduce ContScout, a precise tool for eliminating foreign sequences from annotated genomes. It achieves high specificity and sensitivity on synthetic benchmark data even when the contaminant is a closely related species, outperforms competing tools, and can distinguish horizontal gene transfer from contamination. A screen of 844 eukaryotic genomes for contamination identified bacteria as the most common source, followed by fungi and plants. Furthermore, we show that contaminants in ancestral genome reconstructions lead to erroneous early origins of genes and inflate gene loss rates, leading to a false notion of complex ancestral genomes. Taken together, we offer here a tool for sensitive removal of foreign proteins, identify and remove contaminants from diverse eukaryotic genomes and evaluate their impact on phylogenomic analyses.

Recent technological advances in high-throughput sequencing and plummeting sequencing costs are leading to unprecedented growth in genomic sequence databases[1,2]. Instruments that deliver long-read sequences and the greatly improved throughput of short-read platforms are enabling the resolution of complex eukaryotic genomes in addition to the prokaryotic ones that dominated early sequencing projects. Recently, several large-scale eukaryote sequencing initiatives have been launched with the goal of capturing the genomes of tens of thousands of insects[3], vertebrates[4], fungi[5], plants[6], or ultimately the entire eukaryotic biodiversity on Earth[7].

Due to various biological or technical issues, genomes may contain sequences that do not belong to the targeted organism[8,9] with projects relying on preserved museum- or metagenomic samples are particularly vulnerable to contamination[10–12]. If not carefully addressed, contaminated reference genomes poison public databases with inaccurately labeled sequence data, as demonstrated by a recent study that identified over 2 million records corresponding to contamination in GenBank alone[13]. The extent of contamination within a genome can vary from project to project, but in some extreme cases a separate draft genome representing the contaminant organism could be assembled in addition to the one of the targeted specimen[14,15].

It is well known that contamination can interfere with downstream analyses, be misinterpreted as horizontal gene transfer (HGT)[16,17] and can negatively affect phylogenetic tree inference[18–20]. However, the sensitivity of other phylo- and evolutionary genomic approaches, in particular ancestral genome reconstruction, to contamination has not been explored, despite remarkable expansion of these fields[21,22]. A sensitivity of these approaches to certain analytical issues and HGT has been documented (e.g. Pett et al.[23], Hahn[24]), but the patterns introduced by contamination are poorly known. Because HGT can make genes appear older than they really are[25], and contamination is picked up by orthology assignments, it follows that contamination might have a similar effect to horizontally transferred genes, although, this has not been examined and quantified.

In the last decade, several tools were developed to detect contamination, based on a range of search logics, such as database-dependent taxonomic classification of raw reads or genes using BLAST searches or other similarity-based approaches, utilizing either pre-

---

[1]Synthetic and Systems Biology Unit, HUN-REN Biological Research Centre, Szeged, Szeged 6726, Hungary. [2]U.S. Department of Energy Joint Genome Institute, Lawrence Berkeley National Laboratory, Berkeley, CA 94720, USA. [3]Department of Plant and Microbial Biology, University of California Berkeley, Berkeley, CA 94720, USA. [4]Doctoral School of Biology, Faculty of Science and Informatics, University of Szeged, Szeged 6720, Hungary. ✉e-mail: lnagy@fungenomelab.com

selected marker genes or genome-wide catalogs. However, all these approaches have limitations that preclude their use for explicit tagging and removal of contaminating genes/proteins. Tools that build on selected universal single-copy genes (e.g. CheckM[26], BUSCO[27] and ConFindr[28]) can accurately detect the presence of contamination and estimate its extent, but cannot identify and remove all alien sequences. Most contamination assessment tools, such as CheckM[26], CLARK[29], ConFindr[28] Anvi'o[30] and GUNC[31] focus exclusively on prokaryotes (Archaea, Bacteria) or accept only DNA sequences as input (e.g. Kraken[32], ProDeGe[33], BlobTools[34], PhylOligo[35], CroCo[36], CONSULT[37]). Because DNA evolves faster than protein sequence[38], tools that use the former implicitly assume that the contaminating organism, or its close relative is present in reference databases. This is often not the case even in the best-sampled organismal groups, suggesting that protein-based solutions may be necessary. BASTA[39], Physeter[9] and Conterminator[13] can use protein sequences as input, and the latter was used to flag over 2.1 million DNA sequences in RefSeq and ~14,000 proteins as contamination in the NR database[13]. Despite these developments, efficient and highly sensitive tools that can flag and remove contaminating proteins from genomes and public databases are currently sparse.

In this work, we present ContScout, a tool to identify and remove contaminating proteins from annotated genomes. ContScout assigns a label for each query sequence at six taxon levels of increasing resolution (superkingdom, kingdom, phylum, class, order, family), together with a confidence score. This information is then combined with gene position data, resulting in an improved classification accuracy when compared to existing methods. Screening 844 published eukaryotic genomes with ContScout, we identified 51,222 contaminating sequences primarily from bacteria but also from fungi, plants and metazoans. We also show that, while accurately identifying contaminant sequences, ContScout in most cases does not recognise HGT as contamination. We demonstrate the adverse effects of contamination on evolutionary genomic analyses that lead to spurious ancestral gene count estimates and severely inflated gene loss rates.

## Results

### Description of the ContScout algorithm

We developed ContScout, a contamination detection and removal tool that combines reference database-based taxonomic classification of proteins with gene position data (Fig. 1). Each predicted protein from a query genome is first classified at multiple taxonomic levels via a speed-optimized protein sequence search against a taxonomy-aware reference database using either DIAMOND[40] or MMseqs[41]. Hit lists with taxonomic labels are then trimmed retaining only the top-scoring hits in the first taxon (Fig. 1b), resulting in protein-level taxon labels at all levels of the taxonomic hierarchy (family to kingdom). In the next step, in order to increase sensitivity and specificity, classifications are then combined with contig / scaffold information, leading to consensus taxonomic labels for each contig / scaffold in the assembly (Fig. 1c).

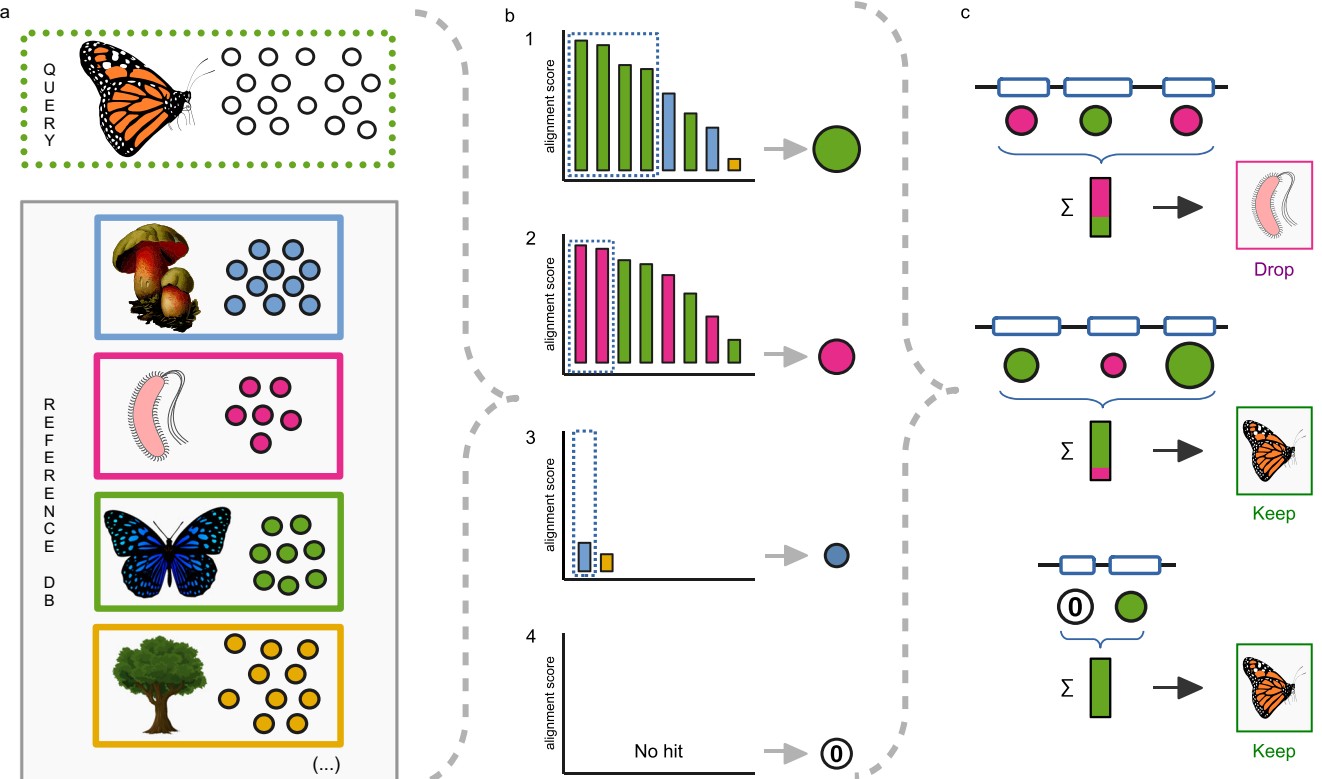

**Fig. 1 | Overview of the ContScout algorithm. a** A quick database search with the query proteins is performed against a taxonomy-aware reference database. The circles represent individual proteins whose color correspond to different taxonomic lineages: green=metazoa, blue=fungi, purple=bacteria, orange=viridiplantae. The expected lineage one of the query genome (metazoa) is shown as a dotted green frame. Each colored frame with a group of colored dots and a thumbnail image represents one of the many reference genomes in the database. **b** The bar charts, illustrating cases 1-4, show query versus reference database alignment scores ranked in decreasing order. Taxon information of the best hit is assigned to each query protein together with a confidence score (proportional to dot size). Protein-wise taxon call examples: Case 1: many hits support the metazoa (green) taxon label that is assigned with a high confidence score. Case 2: bacteria (purple) taxon label is assigned but due to limited support, the confidence score is lower. Case 3: Fungi label (blue) from a single hit is assigned albeit with a very low confidence score. Case 4: No hit observed for query in the reference database. **c** Protein taxon votes are summarized over contigs / scaffolds (∑ sign) and turned into consensus contig calls based on the user-defined threshold. When the consensus taxon label of a contig / scaffold disagrees with that of the query genome, the contig is removed, together with all associated proteins.

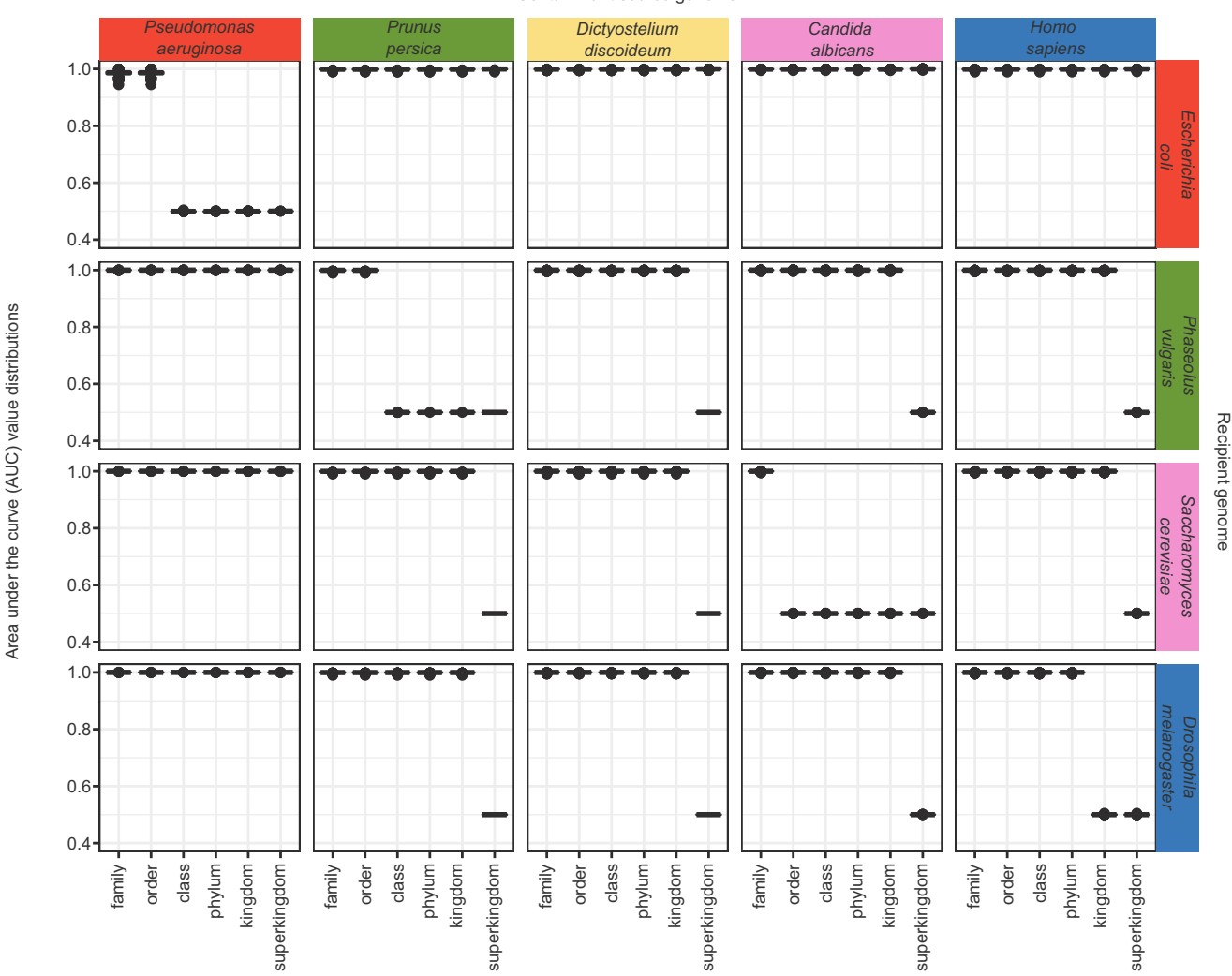

**Fig. 2 | Performance of ContScout on synthetic data.** Artificially contaminated genomes were created by transferring varying numbers of proteins between all possible combinations of source and target proteomes. Proteins were then classified by ContScout as either contamination or host. Matrix of boxplots shows distributions of the calculated area under the curve (AUC) values where column position of charts corresponds to the contamination source genome while row position of charts corresponds to the contamination source genome while row positions indicate the recipient genome. Within each of the boxplots, axis x refers to the taxonomic rank at which decontamination was performed. At each rank, 100 independent ContScout runs were carried out, each of them with 400 randomly selected source proteins being spiked in. See Supporting Fig. 1 for all genome combinations and all spike-in levels located under https://doi.org/10.6084/m9. figshare.23507517. Source data are provided as a Source Data file.

Contigs with the majority of taxonomic labels matching that of the query proteome are kept while those that disagree are marked as contamination and are removed with all encoded proteins (Fig. 1c).

The data storage footprint of ContScout is between 0.1-7,8 GBytes per query genome with a run time of 46–113 minutes benchmarked on a server machine using 24 CPU cores with the RAM usage being constrained to 150 GB. The rate-limiting step is the similarity search that accounts for 80–99% of the total run time (Supplementary Fig. 1).

ContScout is implemented in R, with all software components and their dependencies placed in a Docker container for easy deployment. The software package contains a database downloader script that allows for convenient download and pre-formatting of public reference databases while it also enables users to import custom reference databases of their own.

### Performance assessment on synthetic data

To assess the performance of ContScout, we created pairwise combinations of nine contamination-free genomes (Supplementary Data 1). In each synthetic mix, 100, 200, 400, 800, 1600 or 3200 contaminant proteins were inserted into the recipient proteome and then classified by ContScout as either host or contaminant at several taxon ranks. The genomes were selected so that both distantly related (for example: human in bean) and closely related genome pairs (for example: *Candida albicans* in *Saccharomyces cerevisiae*) could be evaluated. In addition, eight single-direction, biologically inspired synthetic mixtures (i.e. frequently coexisting species) were generated, such as the mosquito genome contaminated with malaria parasites (*Plasmodium*), an insect genome contaminated with a parasitic wasp, or the mouse genome contaminated with human sequences, to name a few (for a complete list see Supplementary Data 1). These data were used to assess the performance of ContScout in classification at the phylum, class, order or family level.

Figure 2 shows area under the curve (AUC) metrics for representative pairwise genome combinations (for all combinations, see Supporting Fig. 1 with the DOI link given at the Data Availability section below). ContScout was able to accurately separate all but one synthetic mixture at the highest taxon rank where the lineages of the two mixed genomes first diverged, with AUC values close to 1. When tested with closely related synthetic mixtures of bacteria (*Pseudomonas aeruginosa*→*Escherichia coli*) and yeasts (*C. albicans* → *S. cerevisiae*),

a

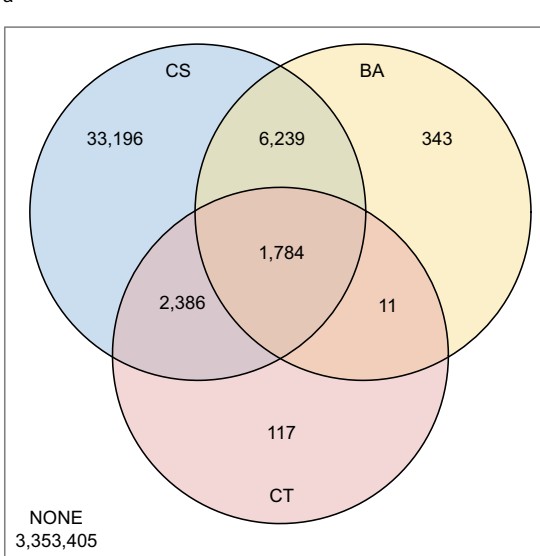

b

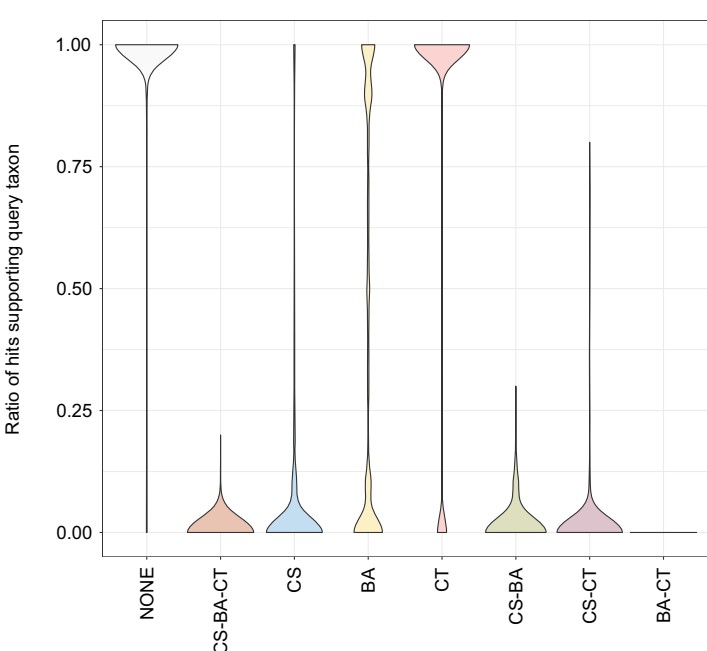

**Fig. 3 | Performance comparison between ContScout, Conterminator and BASTA.** Proteins from the two hundred most contaminated genomes were assigned into eight categories according to the tools that detected them as contaminants. Venn diagram (**a**) shows the number of proteins in each detection category. Letters are as follows: CS: Detected by ContScout, CT: Detected by Conterminator, BA: Detected by BASTA, NONE: Detected by none of the tools. For each query sequence, a taxonomy support value was calculated based on the top 10 hits from the taxonomy-aware UniRef100 database. Violin plots (**b**) summarize taxonomy support ratio distributions within each protein category where value one means perfect support from queries while zero means complete disagreement between the taxonomy label of the query and that of its top hits. Color coding of the violin plots, as well as the letter combinations used in their x axis labels correspond to the different areas of the Venn diagram. Source data are provided as a Source Data file.

ContScout accurately separated contaminant and recipient proteins at the order or family level (Fig. 2) with the AUC range for the order-level bacterial mix being 0.994–0.999, while the family-level AUC range for the yeasts ranging between 0.995-1. In addition, we measured similarly high classification performance on the seven biologically inspired contamination scenarios (Supplementary Fig. 2).

As an outlier among the tested pairs, *Acanthamoeba castellanii* mixed with *Homo sapiens* represented a very difficult case to resolve (Supplementary Fig. 3). Even at the kingdom level, ContScout failed to accurately identify all *A. castellanii* proteins as contaminants, resulting in an AUC range between 0.5 and 0.727. The observed weak classification performance can be attributed to the fact that over 45% of *A. castellanii* proteins did not have enough closely related sequences in the Uniref100 database, making many *A. castellanii* proteins and their associated contigs unclassifiable. By default, such uncharacterized contigs are retained by ContScout that allowed a large proportion of amoeba sequences to escape detection and removal. This known limitation is discussed in more detail, along with analysis options and best practices, in the user documentation that is provided together with the software source code at the GitHub repository (see URL at Code availability section below).

### Comparison of ContScout, conterminator and BASTA

The two hundred most contaminated genomes from the comprehensive eukaryote genome data set (Supplementary Data 3) were used as a benchmark set to compare the detection performance of ContScout with Conterminator (Steinegger & Salzberg, 2020) and the LCA-based tool BASTA[39] (Fig. 3). Among the 3,397,481 tested proteins, ContScout marked 43,605 for removal, while Conterminator and BASTA identified 4298 and 8377 alien proteins, respectively. Hit lists of the three tools overlapped with each other, as 96% and 97% of the proteins tagged by Conterminator and

BASTA, respectively, were identified as alien sequences by at least one additional tool (Fig. 3a). We found that top10 taxon support value distributions (defined as the ratio of matches supporting query taxon among the ten best scoring database hits, see Methods) also agreed well with alien hit lists as 99–100% of the proteins that were tagged by at least two independent tools had a taxon support ratio smaller than 0.25 (Fig. 3b). On the other hand, more than 95% of the 3,353,405 sequences that were considered as non-contaminant proteins by all three tools had a taxon support rate larger than 0.75 (Fig. 3b). Remarkably, out of the 33,196 proteins that were detected exclusively by ContScout, 93.4% had a taxon support rate below 0.25 and only 2% showed a support rate value above 0.75 (Fig. 3b), suggesting that these are accurate contamination calls. At the same time, 33% of the hits exclusively reported by Conterminator and 84% of BASTA-specific hits displayed taxon support values above 0.75 indicating possible false positives. In general, our data suggests that ContScout outperforms both BASTA and Conterminator by accurately identifying five to ten times more contaminating proteins.

### Accuracy assessment on manually filtered genomes

Four sets of manually curated contaminant sequences were collected as ground truth to assess the sensitivity and specificity of ContScout as well as other tools capable of contamination detection (Conterminator) or LCA-based taxon assignment (BASTA, MMSeqs, DIAMOND). Contaminant sequences from *Aspergillus zonatus* (filamentous fungus, $n = 1476$)[42] *Papilio xuthus* (butterfly, $n = 2527$)[43] and, *Bombus impatiens* (bumblebee, $n = 680$)[15,44] genomes were manually curated by the original authors. Additionally, we applied a BUSCO-based protein selection strategy (see Materials and Methods) on the *Quercus suber* (cork oak) genome[45] that yielded 7955 fungal-specific proteins.

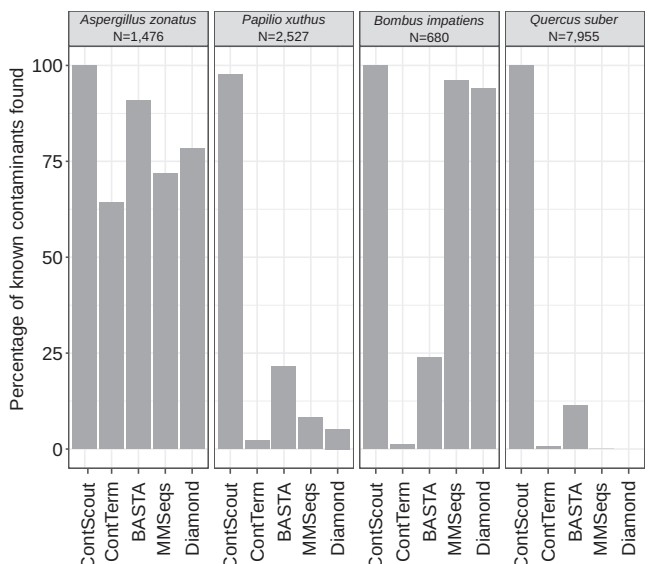

**Fig. 4 | Sensitivity comparison between ContScout, Conterminator (ConTerm), Diamond, MMSeqs and BASTA on manually validated contaminant lists.** Bar charts show the percentage of known contaminants found by each of the tested tools.

While ContScout accurately marked all the 1476 (100%) manually confirmed bacterial proteins in *A. zonatus*, BASTA identified 1341 (91%) contaminant proteins, DIAMOND tagged 1155 (78%), MMSeqs identified 1059 (72%), while Conterminator marked only 948 (64%) (Fig. 4). None of the tested tools yielded any false positives in this genome (Table 1). Similarly, out of the 680 bacterial symbiont proteins that were manually identified as contamination in the bumblebee genome, ContScout precisely identified all 680 sequences (100%), followed by MMSeqs (654 sequences, 96%) and Diamond (639 sequences, 94%). BASTA and Conterminator performed much worse than the other tools tagging only 162 (24%) and 8 (1%) proteins as bacteria, respectively (Fig. 4).

For *P. xuthus*, ContScout identified 2467 (98%) microsporidian proteins, followed by BASTA (542 sequences, 21%) and MMSeqs (207 sequences, 8%). For this genome, Diamond and Conterminator performed worse identifying only 132 (5%) and 57 (2%) of the contaminant proteins, respectively. From this genome, ContScout made one incorrect (false positive) protein call, while no other tools yielded any false positive hit.

Finally, ContScout accurately detected all 7955 fungal proteins in the cork oak genome, while BASTA, the tool with the second closest match, could only identify 909 sequences as fungal (11%). Conterminator and MMSeqs detected 46 (<1%) and 13 (<1%) sequences, respectively, while Diamond failed to accurately call any of the tested proteins (Fig. 4). We also compared ContScout to the DNA-based tool FCS-GX[46]. The two tools performed comparably, removing all bacterial contigs from *A. zonatus* and tagging most of the Microsporidia contamination in *P. xuthus*. Likewise, predictions for *Q. suber* were highly concordant between the two tools with ContScout yielding slightly more hits (Supplementary Table 1).

Overall, our data show that only ContScout managed to detect all manually flagged foreign proteins in the four data sets while most other tested tools missed a considerable proportion of the test proteins. The two genomes in which all contaminant proteins were manually flagged (*A. zonatus* and *P. xuthus*) allowed the assessment of both sensitivity and specificity of the tested tools. Only one protein reported by ContScout for *P. xuthus* turned out to be a false positive, while all other tools reported zero false positives for this genome. None of the tested tools reported any false positive for *A. zonatus* (Table 1).

**Table 1 | Sensitivity and specificity comparison between ContScout, Conterminator, Diamond, MMSeqs and BASTA on manually decontaminated genomes**

| Tool | Genome | True positives | False negatives | False positives |
|------|--------|----------------|-----------------|-----------------|
| ContScout | *A. zonatus* | 1476 | 0 | 0 |
| Conterminator | *A. zonatus* | 948 | 528 | 0 |
| BASTA | *A. zonatus* | 1341 | 135 | 0 |
| MMSeqs | *A. zonatus* | 1059 | 417 | 0 |
| Diamond | *A. zonatus* | 1155 | 321 | 0 |
| ContScout | *P. xuthus* | 2467 | 60 | 1 |
| Conterminator | *P. xuthus* | 57 | 2470 | 0 |
| BASTA | *P. xuthus* | 542 | 1985 | 0 |
| MMSeqs | *P. xuthus* | 207 | 2320 | 0 |
| Diamond | *P. xuthus* | 132 | 2395 | 0 |

### ContScout does not recognize HGT as contamination

To assess how well ContScout can distinguish horizontally transferred genes from contamination, we analyzed HGT events reported from bacteria to anaerobic rumen fungi[47], Ascomycota fungi to Basidiomycota fungi[48] and ten documented bacterial HGTs of antibiotic resistance markers[49]. These data comprised 18 genomes with 1-165 literature-reported HGT genes. Table 2 shows that ContScout did not recognize any of the proteins encoded by HT genes in eukaryote genomes as contamination, with the exception of 1 out of 79 (1.2%) in the case of *Armillaria ostoyae*, 1 out of 165 (0.6%) in *Neocallimastix californiae* and 6 out of 117 (5.1%) in *Orpinomyces* sp. Similarly, ContScout removed only one antibiotic resistance marker of known HGT from *Phascolarctobacterium succinatutens* while it kept all proteins encoded by HGT genes in the other nine tested bacterial genomes (Supplementary Data 2).

Upon scrutinizing CountScout's decision process, we can see that a larger number of proteins were labeled as suspicious in the protein-wise taxonomic tagging step of the algorithm (Table 2 and Supplementary Data 2); however, the majority of these were later confirmed as host proteins in the second step, when taxonomic information is summed over contigs.

Overall, ContScout flagged 0–5.1% of previously published HGT genes in eukaryotes as contamination, indicating that in the context of these examples, it could distinguish these two types of alien proteins in annotated genomes. Similarly, in nine out of ten tested cases, ContScout did not confuse bacterial HGT events for contamination (Supplementary Data 2). Notably, a close inspection of the removed bacterial sequence revealed that it most likely represents a 3.8 kb circular plasmid. Since all five encoded proteins on the plasmid were tagged as alien, the whole plasmid was subsequently removed by ContScout. The risk of confusing non-integrated transferrable extrachromosomal elements with contamination is a known limitation of ContScout that is further discussed below in the Discussion section.

### Rampant and diverse contamination in eukaryotic genomes

To assess levels of contamination across high-level taxa (kingdoms, supergroups) in public genome databases, we analyzed ContScout outputs obtained on a set of 844 eukaryotic genomes representing all major lineages (341 metazoans, 129 plants, 272 fungi and 102 other eukaryotes, Supplementary Data 3,4). ContScout revealed the presence of rampant contamination, detecting at least one contaminating protein in 447 genomes, on average reporting 114 alien proteins per genome (range: 1–12,656, see Fig. 5a). The prevalence of contamination was lowest among fungi (43% of the species), slightly higher in animals (55%) and plants (56%) and highest among other eukaryotes (66%).

**Table 2 | The performance of ContScout on literature-reported cases of horizontal gene transfer**

| Species | # of reported HGT genes | HGT type | # proteins flagged as suspicious in step 1. | # HGT proteins discarded as contamination |
|---|---|---|---|---|
| *Armillaria cepistipes* | 66 | Fungi to fungi | 2 | 0 |
| *Armillaria mellea* | 77 | Fungi to fungi | 4 | 0 |
| *Armillaria ostoyae* | 79 | Fungi to fungi | 3 | 1 |
| *Armillaria solidipes* | 73 | Fungi to fungi | 2 | 0 |
| *Anaeromyces sp.* | 147 | Bacteria to fungi | 4 | 0 |
| *Piromyces finnis* | 132 | Bacteria to fungi | 3 | 0 |
| *Neocallimastix californiae* | 165 | Bacteria to fungi | 12 | 1 |
| *Orpinomyces sp.* | 117 | Bacteria to fungi | 6 | 6 |

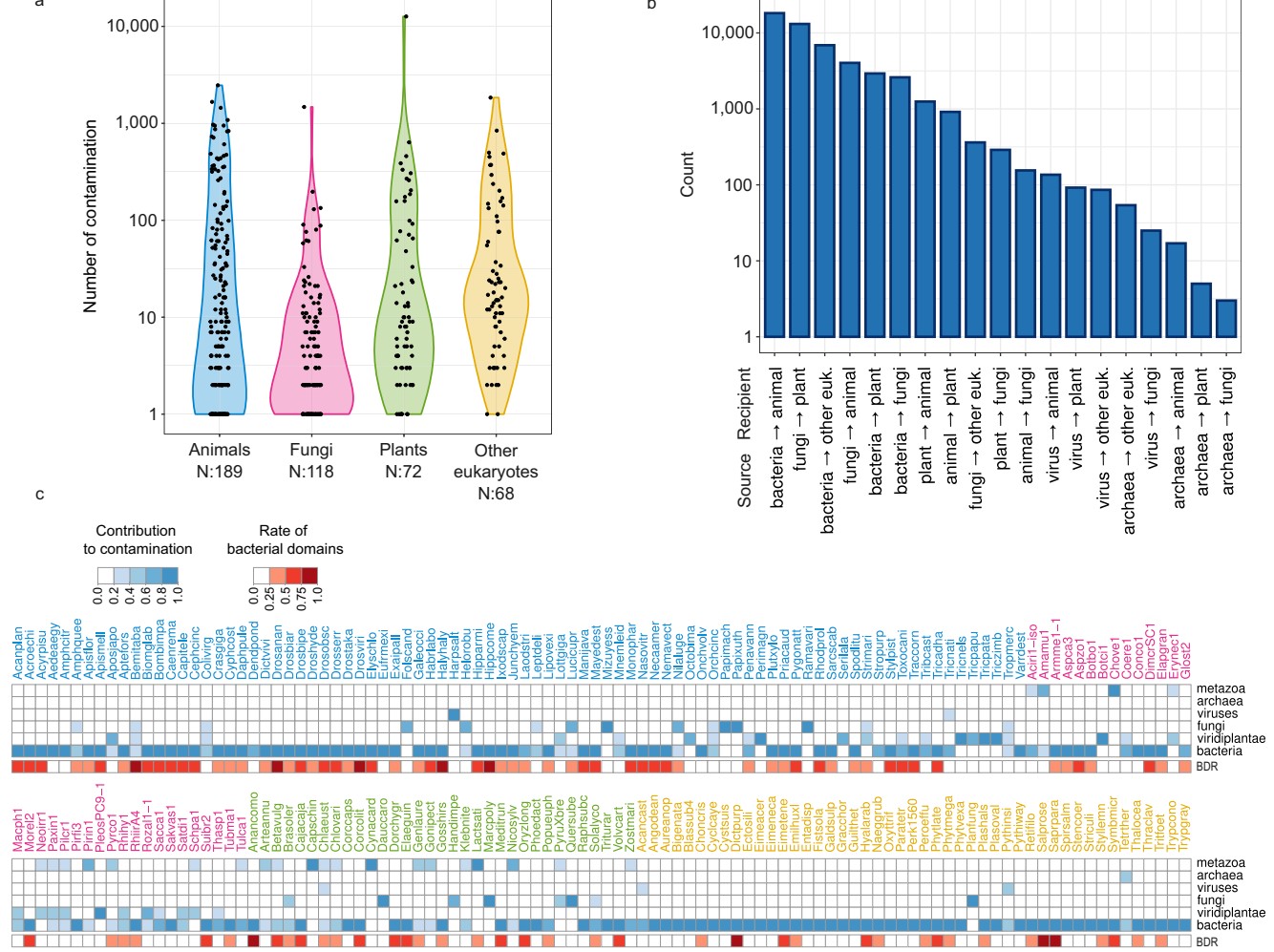

**Fig. 5 | Summary of contamination statistics across 844 genomes. a** Violin plot showing the number of contaminant proteins detected in 844 eukaryote genomes. Proteomes with no contamination (N: 397) were omitted from the plot. Violin plots are colored according to the taxonomic lineages of the tested genomes with blue color representing animals, purple corresponding to fungi, green standing for plants and orange for other eukaryotes. **b** Bar plot summarizing the numbers of proteins between each detected contamination-recipient pair. Pairs are charted in decreasing order. **c** Heatmap (cells in shades of blue) indicate the contributions of each high-level taxa to contaminants detected in each of the top 200 contaminated genomes. Heatmap label colorings matches the color coding used for the violin plots in panel a, as described in the figure legend above. For full-length species names, see Supplementary Data 3. BDR (cells in shades of red) corresponds to the ratio of domains of bacterial origin among the domains detected on contaminant proteins. Source data are provided as a Source Data file.

Bacteria and fungi turned out to be the most frequent sources of contamination, donating 30,666 and 17,531 alien proteins, respectively. Of the fungal proteins, however, 12,631 could be linked to a single massively contaminated plant genome *Q. suber* (Fig. 5b).

Viridiplantae (1538) and Metazoa (1069) together accounted for no more than five percent of the contaminating proteins. Viruses yielded 273 contaminants while 76 contaminating proteins with an archaeal origin were detected.

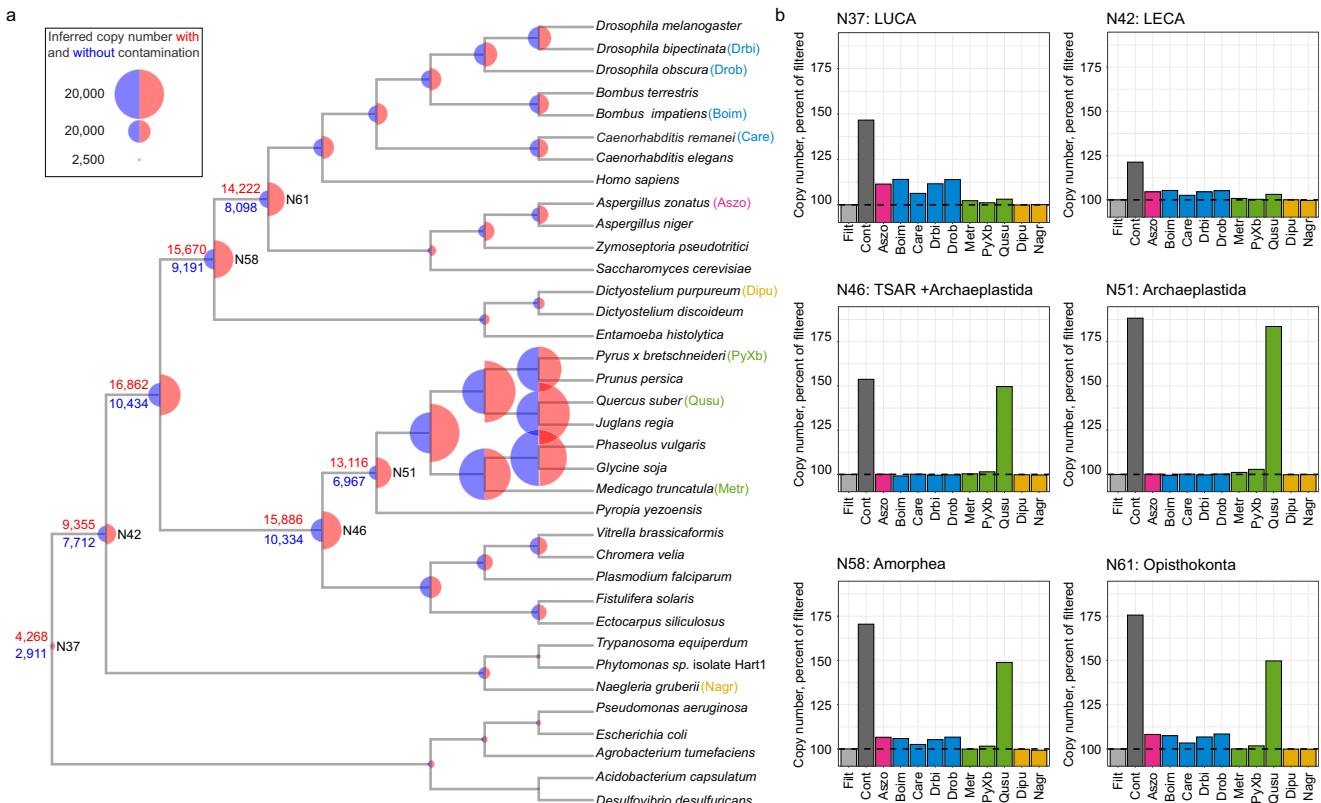

**Fig. 6 | Effect of contamination on ancestral genome reconstruction in the 36 genome data set. a** Proportionally scaled, color-coded semi-circles correspond to copy number estimates, calculated by Compare[22] with the red color representing contaminated and blue color standing for cleaned data. Additionally, the copy numbers for seven prominent internal nodes are provided within the species tree as text node labels. **b** Bar plots show the bias of individual contaminated genomes introduced to copy number prediction at six selected internal nodes (N37: LUCA, N42: LECA, N46: TSAR+Archaeplastida, N51: Archaeplastida, N58:Amorphea,

N61:Opisthokonta), where 100% corresponds to the copy number inferred from clean data. Abbreviations used on bar plot legends: Filt: decontaminated, Cont: contaminated. Species name abbreviation key is provided as part of the species tree leaf node labels in panel a. Bar plot are colored according to taxon lineages of the genomes with blue color representing animals, purple corresponding to fungi, green standing for plants and orange for other eukaryotes. Bars corresponding to clean data are colored in light grey, while those ones that stand for unfiltered data are charted in dark grey.

Of the 200 most contaminated genomes, 140 had contaminants originating from multiple sources, while in 60 cases contaminants could be traced to a single high-level taxon source. Out of these, bacteria turned out to be the sole source of contamination in 55 cases (Fig. 5c). The prevalence of bacteria as a source of contamination was also confirmed by Pfam domain analysis within the contaminant sequences: in 52 out of the 200 most contaminated genomes, Pfam domains exclusive to bacteria made up more than 50% of the surplus domains that were not truly part of the query genome but got assigned due to contamination. Sequences marked by ContScout are listed in Supplementary Data 4.

Since ContScout applies a consensus taxon calling over groups of proteins that are encoded on the same contig, it can occasionally remove proteins that, based on their protein-wise taxon calls match that of the query. Using the 200 most contaminated genomes, we counted the number of proteins that matched the (high-level) taxon of the query but were removed due to their contig contexts. We found that the number of such proteins remained low in all tested cases (Supplementary Data 5).

### Contamination bias analyses of ancestral gene content

We next addressed the hypothesis that contamination can bias phylogenomic analyses of gene content and reconstructions of ancestral genomes. Both approaches have recently gained momentum and, in several taxa revealed an early burst of gene duplication followed by gene loss, which was postulated to be a dominant mechanism of

genome evolution (e.g., refs. 50–53). However, how contamination influences ancestral genome reconstruction has not been assessed to date.

We addressed this question here using reconstructions of ancestral gene content in early eukaryote ancestors in a 36-species data (Supplementary Data 6) set containing 10 contaminated genomes and inferred gene gain, duplication and loss patterns using the original genomes and those cleaned by ContScout. Figure 6 shows considerable differences in inferred ancestral genome sizes between contaminated and clean data. For example, based on contaminated data, Last Eukaryotic Common Ancestor (LECA) possessed 9355 protein-coding genes, whereas in decontaminated data it possessed 7712, a 21% overestimation. The largest difference (88% overestimation) was observed in the Archaeplastida most recent common ancestor (MRCA, N51 on Fig. 6), in which contaminated data suggested 6149 more ancestral genes (13,116) than decontaminated data did (6967). This node reflects a single strong signal from the heavily contaminated *Q. suber* genome, including 12,631 fungal genes. Using a series of partially decontaminated data, we measured the effect of each of the contaminated genomes. As expected, ancient gene copy number overestimation at the MRCA of Archaeplastida (N51) was mostly explained by contaminations in *Q. suber*, while at Last Universal Common Ancestor (LUCA) of all life (N37) it mainly originated from multiple insect and fungal genomes contaminated with bacterial sequences. These results reveal that the effect of contamination additively builds up as we move from recent to ancient nodes in the tree.

Across the whole data set, the contaminated analysis suggested 11,062 more gene gains and 63,243 more gene losses than the decontaminated analyses, indicating that contamination-biased gene losses nearly five times more. To uncover why contamination inflates gene loss to such a great extent, we manually checked the pyridoxal kinase protein family that is involved in the pyridoxal 5'-phosphate salvage pathway. This family is conserved across the entire tree of life and ContScout identified one contaminating protein in *Q. suber* and one in *B. impatiens* (Supplementary Fig. 4). The maximum likelihood gene tree readily identified two mis-positioned proteins: one of the *Q. suber* proteins clustered with fungi (Quersube_4764, SH support value: 0.98), whereas one from *B. impatiens* grouped in the bacteria (Bombimpa_11962, SH support value: 0.97). In line with the gene tree, ContScout tagged Quersube_4764 as fungal and Bombimpa_11962 as bacterial.

Gene tree - species tree reconciliation and mapping of gene gain/loss events on the species phylogeny indicated that the contaminated and decontaminated gene trees could be explained by 23 (6 gene gains and 17 losses) and 9 (7 gains, 2 losses) events, respectively (Supplementary Fig. 5). We found that Quersube_4764 and Bombimpa_11962 have induced eight and ten gene losses, respectively. In the case of Quersube_4764 these losses were introduced because, during the mapping, it was assigned to a 1-to-1 orthogroup with a fungal protein (Zymps_805618) the origin of which was mapped to the most recent common ancestor of plants and fungi. It follows that for this orthogroup to be explained along the phylogeny, losses had to be counted for all descendents of the plant/fungal ancestor except *Q. suber* and *Zymoseptoria tritici*.

Taken together, these results indicate that contamination can introduce considerable bias into ancestral genome reconstruction and uncover how it inflates gene loss estimates in particular.

## Discussion

In this paper, we presented a tool for identifying contaminating proteins in annotated genome sequences and demonstrated that in evolutionary genomics contamination can lead to a false notion of complex ancestral genomes and overestimated gene loss rates. Contamination is a widely recognized problem in sequence databases and can stem from a variety of reasons (reviewed by Cornet et al.[54]). Several tools have been developed for the detection of contamination in large sequence databases[13] or estimating contamination level in (meta)genomes (e.g. CheckM[26], BUSCO[27]). Most previous tools focus on classifying raw sequencing reads as host or alien[29,32] or rely on measures of similarity to a pre-selected set of marker genes[26,27,55], whereas genome-wide tools that can clean genomes of contamination are at paucity[13]. ContScout is a genome-wide method that relies on a reference database and genome annotation data to identify and remove contaminating proteins. After inferring protein-wise taxon calls based on similarity searches against taxonomy-aware reference databases, ContScout summarizes these across contigs/scaffolds and flags contaminating sequences. Thus, ContScout can clean genomes from encoded contaminating proteins. We anticipate this feature will become more and more important as genome sequencing efforts are extended to field and museum specimens, mixtures of organisms (e.g. host and its parasite, metagenomes) or unculturable single-cells, all of which increase the risk of contamination. Our analyses of synthetic data, benchmarking against manually curated sequences, as well as a comparison to other decontamination tools indicated that ContScout achieved high sensitivity and specificity even at fine taxon levels (family, order), while it spared horizontally transferred genes and outperformed most competing tools. The only exception where ContScout achieved lower accuracy were protist genomes, which can be explained by the scarcity of sequenced genomes in these taxa, a situation that is expected to change quickly. Depending on the genome, ContScout required 46-113 minutes on a server computer.

We think the overall good performance of our tool is rooted in the combination of taxonomic classification with scaffold-level decision making. First, the non-fixed size of considered hit lists may yield more robust taxon call assignment than fixed-size hit lists (e.g. top 100 hits), which are used in most previous software (see MEGAN[56] for exception), and can help minimize the effect of mis-labeled proteins, sporadically present in reference databases. Second, if a contig is marked for removal by ContScout, any ambiguous proteins coded on it will be also discarded, which potentially increases sensitivity. On the other hand, those horizontally acquired genes that are integrated into the chromosome of the recipient organism, are not discarded by ContScout, because most proteins encoded on those chromosomes/scaffolds will match the taxon label of the query genome. However, when the foreign DNA does not integrate into a recipient chromosome but remains maintained as a circular plasmid, ContScout will identify it as contamination as seen with the *SAT-4* streptothricin acetyltransferase region in *P. succinatutens* (See Supplementary Data 2). This is a known limitation in ContScout's ability to handle certain HGT events. Additional limitations of ContScout probably lie in fragmented genome assemblies (i.e. low N50 values, small contigs), screening within undersampled groups (e.g. protists) and in chimeric contigs, the latter of which albeit exist, are likely rare[57,58].

Our analyses of literature-reported HGT events demonstrate that, in the context of the analyzed empirical examples, ContScout performed well in distinguishing contamination from both recent and ancient HGT. This is an important ability since both contamination and HGT can display signals of alienness relative to the host genome, but removing HGT is not a goal of most decontamination analyses. We found that ContScout greatly outperformed Conterminator and BASTA, two recent cleaning tools for protein data. For example, while ContScout identified all proteins that were manually flagged as contamination in *A. zonatus*[42], Conterminator and BASTA identified only 64% and 91% of them, respectively. We hypothesize that the loss sensitivity of both BASTA and Conterminator lies in using fixed similarity thresholds with values being set high by default. This implicitly assumes that the contaminating organism, or its close relative, is present in the reference database. Even with the dense sampling of genomes we have today, having the genome of the exact contaminating taxon in the database is a rare situation, so we think the more dynamic and sensitive search engine implemented in ContScout is warranted. Comparison of ContScout with FCS-GX[46] revealed that the two tools yield highly concordant results (Supplementary Table 1) with FCS operating exclusively at the DNA level and ContScout dealing with predicted proteins backed up by protein annotation data.

Using ContScout we screened 844 published eukaryotic genomes and found widespread contamination, most commonly bacteria and fungi. We identified >50,000 contaminating proteins in this set, while Conterminator identified 327–14,148 depending on the database configuration used. These figures agree with previous reports of contamination in reference sequence databases[59] and (meta)genomes[10,26,60–62], however, our inventory highlighted a range of novel patterns. First, the number of contaminating proteins covered three orders of magnitudes: it ranged from a handful of proteins up to >12,000, in extreme cases allowing the subtraction of presumably complete protein repertoires of the contaminating organism[14,15] from the contaminated genome. The taxonomic distribution of contaminating organisms reflects common lifestyles of microbes as symbionts, parasites, food sources or commensals. Previous studies have also reported bacteria as a common contaminant[8], whereas in our analyses fungi also emerged as frequent contaminating organisms, possibly due to their diverse associations with plants and metazoans. Finally, we expect the cleaned genome annotations for 844 eukaryotes, covering all eukaryotic supergroups and phyla to form a gold standard resource on which comparative analyses can be built.

Whereas the effects of contamination on metagenomic studies and functional interpretation of genomes is quite straightforward[31,57], the biases they cause in the context of evolutionary genomics is poorly explored. We found that ancestral genome estimation can suffer when contamination is present in the data: alien proteins pushed gene family origins towards the root of the tree, resulting in an overestimation of ancestral gene contents (up to 80%), and that of the number of gene losses (up to 5,7-fold). This effect was additive when multiple contaminated species were present in the data set. This can give the impression of highly complex ancestral genomes, as inferred in recent empirical studies on ancestral gene content in several groups, such as animals, plants[50] or LECA[63]. A recent report demonstrated that incomplete genome annotation and unrecognized HGT can also inflate gene loss estimates[64]. While genome annotation errors will induce excess losses distributed randomly across the tree and affect mostly terminal branches, excess loss introduced by contamination and HGT might cause more serious problems. Indeed, in our analyses contamination yielded a strong bias in gene loss estimates, mostly affecting deep branches and causing a several-fold overestimation. These results underscore the necessity of assessing contamination levels of genomes before being included in comparative genomic analyses.

In summary, we developed a highly sensitive contamination detection and removal tool, demonstrated its utility for decontaminating large numbers of published and annotated genomes even in the presence of HGT, provided a broad set of cleaned eukaryotic genomes and uncovered a massive impact of contamination on evolutionary genomics studies. Given the widespread occurrence of contamination in the analyzed genomes, we advocate the reporting of measures of contamination in publications of annotated genomes and expect that ContScout and the analyses presented here will facilitate the accumulation of high-quality genomes and improve their utilization in diverse fields.

## Methods
### Selection of a comprehensive eukaryote data set
In total, 844 published genomes, comprised of 341 animals, 272 fungi, 129 plants, and 102 other eukaryotes, were downloaded from public databases (JGI MycoCosm[5], ENSEMBL[65], Genbank[66]) to perform a broad contamination screening. If multiple isoforms of the same gene were present, we selected the longest one for analysis. Genomes included in the study, together with their source databases and accession numbers, are summarized in Supplementary Data 3. Date of data collection: July, 2019.

### Selection of a 36-genome data set
In order to assess the effect of contamination on ancestral genome reconstruction, a 36-genome data set has been compiled encompassing five bacteria, eight animals, four fungi, seven plants and twelve other eukaryotes. Altogether, the data set included ten genomes (*Aspergillus zonatus*, *Bombus impatiens*, *Caenorhabditis remanei*, *Dictyostelium purpureum*, *Drosophila bipectinata*, *Drosophila obscura*, *Medicago truncatula*, *Naegleria gruberii*, *Pyrus x bretschneideri*, *Quercus suber*) that contained between 28 and 12,656 contaminant proteins each. Contaminated genomes, each matched with a contamination-free related genome, were selected so as to represent all major eukaryotic lineages in which we frequently found contamination.

### ContScout run parameters
ContScout runs were carried out using the docker image h836472/contscout:natcomm. Uniref100 database (release 2022_1) was selected as the reference database (-d uniref100) with MMSeqs used as the search engine (-a mmseqs) with the search sensitivity set to very fast (-s 2). The minimum sequence identity threshold was set to 20 percent (-p 20).

### Performance testing on synthetic data
Nine genomes (2 bacteria, 2 animals, 2 fungi, 2 plants, 1 other eukaryotes) from the 844-genome data set with no evidence of contamination (based on analyses of the 844 genomes) were collected and used to assess the contamination / host classification performance of ContScout. Artificially contaminated genomes were created by transferring proteins between each possible source and recipient genome pair within the data set. Additionally, seven genome mixes were created between pairs of closely related genomes taken from the 844 genome set, each mimicking a plausible real-life contamination scenario (amoeba→human, human→mouse, fungi→fungi, nematode→pig, *Plasmodium*→mosquito, wasp→moth, alga→plant, for details see Supplementary Data 1).

For each genome pair, a set of 100, 200, 400, 800, 1600 or 3200 randomly selected proteins were transferred, assigned to random virtual contigs each holding one, two, five, ten or twenty alien proteins. At each of the six spike-in levels, 100 random replicate sets were generated. ContScout was then executed on the artificially contaminated data to classify proteins as either host or contamination. Receiver operating characteristic curves were calculated by the pROC[67] package in R[68], with the area under the curve (AUC) metric being used to assess the classification performance.

### Performance assessment on manually curated genomes
The 844-genome data set included three projects (*A. zonatus*[42], *B. impatiens*[44] and *P. xuthus*[43]) where the authors carried out manual genome decontamination and released clean assembly versions after our data collection took place. In addition, Martinson and co-workers released a separate draft genome for the gut symbiont gamma proteobacterium[15], which they identified as the sole source of contamination within the original *B. impatiens* assembly. For *A. zonatus*, and *P. xuthus* we used the proteins that were removed by authors as a ground truth, yielding 1476 and 2527 manually curated contaminants, respectively. For *B. impatiens* we mapped back proteins from the published symbiont proteome to the contaminated *B. impatiens* data using a sequence identity threshold of 95% and a sequence coverage threshold of 0.6. That way, we identified 680 symbiont proteins within the *B. impatiens* proteome that served as positive control for contamination.

Our preliminary analysis of the 844 genomes indicated a massive fungal contamination in the *Q. suber* genome[45]. In order to validate this finding ascomycota, as well as plant-specific proteins were marked in the draft genome based on a search with BUSCO v5.4.4[27] using taxon-specific orthoDBv10[69] reference HMM sets.

Contigs that contained a minimum of 100 proteins, out of which at least 20 were specific to ascomycota and had no hits specific to plants, were classified as confirmed contaminants of ascomycota origin. A total of 7955 proteins coded on the selected ascomycota-specific contigs were used as a ground truth for contamination in *Quercus*.

### Large-scale comparison between ContScout, Conterminator and BASTA
In order to carry out a Conterminator screen in a way similar to that of[13], the 844-genome data set was combined with the UniRef100 database[70] (release 2022_01). Whenever a protein sequence was present in both sources, redundancy was resolved by keeping only the copy from the 844-genome set. Conterminator was then executed in protein mode, using default parameters. Then, Conterminator hit list was truncated to keep only the subset that corresponded to the 200 most contaminated genome, as identified by the ContScout screen of the 844-genome dataset. Similarly, hit list of the 200 most contaminated genomes of the 844 data set were extracted from ContScout hit list and used in the comparison.

BASTA[39] (V1.4) was used with default parameters to assign taxon labels to each protein of the 200 most contaminated genomes of the

844 genome data set, using the Uniprot database (downloaded by BASTA on the 8th of November 2022) as the reference. Any mismatch between the obtained and expected taxon call at high-level taxon ranks (superkingdom/kingdom) was considered as contaminant.

Hit lists of the three tools were compared grouping query proteins into groups according to the tools that identified them as contaminant. Taxon support value distributions based on the best ten reference hits per query were also calculated and visualized.

### Taxon support ratio calculation

Taxon tags suggested by the ten best-scoring database hits were taken for each query protein. Taxon support ratio was calculated as the ratio of tags supporting the expected query taxon among the ten best hits. Value close to 1 suggests genuine host protein while value close to 0 indicates strong disagreement between the expected and observed taxon data.

### Domain analyses

Interproscan v5.44.79.0[71] was used to search for protein domains in the comprehensive eukaryote genome set. Bacterial-specific domains were extracted from the Pfam database[72] based on the ratio of bacterial sequences within the seed alignments. Domains with over 95% bacterial sequences in their seed alignments were considered as bacterial. In order to collect Fungi-specific domains, the UniprotKB database[73] together with IPR annotations was downloaded. A domain was considered as fungi-specific if at least 95% of the associated UniProtKB proteins originated from the kingdom Fungi.

### Ancient genome reconstruction and copy number estimation

We followed published pipelines for reconstructing ancestral genomes using the G36 dataset (Supplementary Data 6), which briefly, utilized a species tree and reconciled gene trees for each of the protein families identified in the input genomes[22,74,75]. For inferring a species tree, BUSCO[27] v3 HMM profiles were used to collect 428 conserved single-copy candidate proteins from the decontaminated G36 data set. MMSeqs was then applied to calculate an all versus all protein similarity network among the proteins on which Markov-clustering with hipMCL[76] was carried out with an inflation parameter of I = 2 to identify protein families. Predicted protein families were filtered manually, keeping only the conserved single-copy ones. Mafft (v7.407[77]) with the --auto option was used to perform multiple sequence alignment for each single-copy protein family. Uninformative and poorly aligned parts were removed with TrimAl[78] (parameters: -gt 0.95) and the resulting trimmed alignment were concatenated into a supermatrix of 428 protein families and 172,083 characters. RAxML 8.2.12[79] was used to infer a maximum-likelihood species tree under the PROTGAMMALG model of protein evolution. The model was partitioned gene-by-gene.

For assessing the impact of contamination on ancestral genome reconstruction, a series of semi-decontaminated data sets was generated based on the 36-species collection where each dataset retained contamination from only one of the ten contaminated genomes (*Aspergillus zonatus*, *Bombus impatiens*, *Caenorhabditis remanei*, *Dictyostelium purpureum*, *Drosophila bipectinata*, *Drosophila obscura*, *Medicago truncatula*, *Naegleria gruberii*, *Pyrus x bretschneideri*, *Quercus suber*). The data series was then completed with the fully decontaminated as well as the original G36 versions. For each variant in the series, orthologous protein families were identified by Orthofinder v2.4.1[80] using the species tree as a reference. Ancient genome reconstruction as well as gene gain/loss events were inferred by using the COMPARE pipeline as described before. Effects of individual contaminated genomes were determined by comparing gene gain/loss counts between contaminated and clean versions.

### Reporting summary

Further information on research design is available in the Nature Portfolio Reporting Summary linked to this article.

## Data availability

Genomic data used in the study are available in the JGI Mycocosm / ENSEMBL / NCBI databases. Individual source information and accession number for each genome is provided in Supplementary Data 3 and Supplementary Data 6. The Uniref100 database, that was used as a reference, is available at UniProt under the release number 2022_01. All custom R scripts written to perform the analyses within the presented study as well as codes created to summarize and visualize the results have been deposited to the Figshare repository under the https://doi.org/10.6084/m9.figshare.23507517. Source data are provided with this paper.

## Code availability

An executable image of ContScout with the version matching the one used for the study can be downloaded as a Docker image from DockerHub repository under h836472/contscout:natcomm. Source code for ContScout, together with a user manual, tutorials and a minimal example data, is provided at https://github.com/h836472/ContScout/ under the NatComm branch. Third-party software tools that were used for data manipulation, data analysis and visualization are listed in Supplementary Table 2.

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

## Acknowledgements
This work was funded by the Momentum Program of the Hungarian Academy of Sciences (LP2019-13/2019 to LGN) and by the European Research Council (Grant No. 758161, to LGN) (both to LGN). This research was performed under the Facilities Integrating Collaborations for User Science (FICUS) program (proposal: https://doi.org/10.46936/10.25585/60008430) and used resources at the DOE Joint Genome Institute (JGI) (https://ror.org/04xm1d337) and the National Energy Research Scientific Computing Center (NERSC) (https://ror.org/05v3mvq14), which are DOE Office of Science User Facilities operated under Contract No. DE-AC02-05CH11231.

## Author contributions
L.G.N., B.B, Z.M., B.H. and I.G. conceptualized and designed the research, B.B., wrote the code and performed the analyses. B.B, C.F. and Z.H. performed the test of the tool, L.G.N., Z. M. and B.B., interpreted the results and wrote the paper. All authors have read and agreed to the published version of the manuscript.

## Funding

## Competing interests
The authors declare no competing interests.
