## [Peer Review File · Nature Communications]

REVIEWER COMMENTS

Reviewer #1 (Remarks to the Author):

Review of the manuscript „ContScout: a novel tool for accurate genome decontamination with critical impact on the analysis of ancestral genomes” submitted by Balint et al.

In this manuscript, the authors present a novel approach for sequence-similarity based contamination detection in gene sets inferred from genome (draft) sequences. Each protein sequence is searched against UniRef100, and a dynamic threshold is applied to assign the sequence to a higher order taxon, where the considered taxa are at least on the kingdom level. On this basis, they propose to decontaminate assemblies by removing contigs/scaffolds of which at least half of the genes have been flagged as contaminants. The authors further present several analyses to show the impact undetected contaminants can have on downstream evolutionary analyses including the assessment of ancestral gene set sizes as well as the prevalence of lineage-specific gene gains and losses. Overall, the analysis touches on a relevant topic that was addressed by several publications that appeared in the last years. The findings reported here reproduce that genome assemblies in the public domain are, in parts, heavily contaminated. They further support the general understanding that contaminations can be positively misleading in evolutionary analyses that are unaware of their existence.

Major issues

1. Taxonomic assignment of protein sequences is well established in the literature, and it is common to apply a least common ancestor approach (LCA) on a dynamically selected set of Blast/Diamond top hits (see e.g. Huson et al. 2007; DOI: 10.1101/gr.5969107; and an LCA-based taxonomic assignment is also implemented into Diamond). The authors here propose their own rule-based approach considering a dynamically selected set of top hits resulting from a database search with either Diamond or MMSeq2. Since the approaches are conceptually similar--with the difference that the LCA approach integrates the taxonomic assignments of the considered hits instead of going for a majority vote--they should be exhaustively compared. A specific focus should be placed here on the simulations where the rule-based approach presented here did not perform well, e.g. contaminations by *Dictyostelium*.
2. The contamination screen is limited in its resolution because sequences are assigned only to higher-order taxa on the level of kingdoms and beyond. This precludes the identification of contaminations by sequences from the same kingdom or domain. Although the authors state that it is straightforward to increase the resolution of their taxonomic assignment, I consider it essential that this is shown. Essentially, the LCA approach does not share this limitation, and hence the authors would need to show that their rule-based approach implemented into ContScout extends the state-of-the-art in the field.
3. The authors describe the impact that contaminations have on the outcome of evolutionary studies, where they basically result in the overestimation of gene age, gene loss events, and of HGT (which is not considered here). This topic has received considerable attention over the past years (see e.g., Cornet and Baurain 2022). Although it is nice to see numbers on a tree (see Figs. 5 and 6), the corresponding values heavily depend on the precise taxa under study, and with the granularity of the contaminant detection (see above). Thus, what is presented here is hard to generalize, with the exception of the amounts of foreign genes in the individual assemblies. It is necessary to carve the novelty in the present study.
4. How were the 36 taxa selected that formed the data basis for studying the evolutionary implications of contaminations? It is surprising that several taxa, especially within the animals, are highly overrepresented: *Caenorhabditis* – 2 species, *Bombus* – 2 species, *Drosophila* 3 species.
5. On page 11, the authors investigate why contaminations inflate gene loss estimates nearly five times more than gene gain estimates. This finding is conceptually easily interpretable, because a contamination (again similar to a horizontal gene transfer) introduces a foreign gene into one taxon but not into its relatives. If we use a naïve Dollo parsimony-based approach of reconstructing the evolutionary fate of the gene, this will lead to a vast overestimation of the gene age resulting in an inflated number of only gene losses but rarely of gene gains. If the authors plan to keep this analysis in the results, it would be very helpful if they could provide additional motivation on why they think that this finding is important to report.
6. The tree, as it is shown in Figs 5 and 6 reflects the Coelomata-hypothesis (*Drosophila* and

human grouped to the exclusion of *C. elegans* instead of the commonly accepted Ecdysozoa-hypothesis (*Drosophila* and *C. elegans* grouped to the exclusion of humans). If the authors want to keep this phylogeny, they should explain why this is justified.

7. The definition of orthoGroups in the text (line 355-356) reads like 1:1 orthologs where the orthoGroup represents a clique of orthologous proteins (the orthology assignment criterion is fulfilled for each pair of proteins in the orthoGroup, see e.g. OMA groups). This would be a very strict criterion resulting in many but typically very small orthoGroups. More inclusive approaches integrate also inparalogs into the orthologous groups (see OMA hierarchical orthologous groups). The authors should increase the level of precision here. Moreover, the context in which the orthologous groups are mentioned is not clear. The inclusion of a contamination into an orthologous group will always inflate a gene age estimate by Dollo parsimony irrespective of how the group was compiled. In fact, I would expect that more inclusive orthology assignments likely suffer from a stronger impact of contaminations because of a tendency to generate larger orthologous groups. This is contrary to what is stated in the manuscript where the authors mention their expectation that 'simpler' approaches are less sensitive to the distorting effects of contamination. In any case, such speculations should be replaced by an analysis that is backed up by data.

Minor issues

1. What is RLE?

2. The authors state that people use fixed-size hits for taxonomic assignments (e.g. Top100). This is not state of the art (see MEGAN), and should be presented in a more differentiated manner.

3. Caption of Fig. 3 could be condensed

4. The heading 'Phylogenomic analyses are biased by contamination' should be rephrased, since many people connect with 'Phylogenomics' the inference of evolutionary relationships between species using large taxon-gene matrices. This, however, is not in the focus here.

5. What is the rationale of limiting the computation of the taxonomy support value (Fig. 4) to the top ten hits when the taxonomic assignment uses a dynamic approach?

6. In the discussion, the authors mention the application of ContScout in the context of decontamination of genome assemblies by removing human sequences. Removal of contaminating human sequences is routinely done on the read level prior to genome assembly, and rapid kmer-based approaches, such as Kraken2 are perfectly suited for this application since the contaminating species is known, and a finished genome assembly is at hand.

7. HLT should be explained when it occurs first in the text, since this clarifies right from the start the limits in resolution.

Reviewer #2 (Remarks to the Author):

Bálint et al., present a new algorithm for contamination detection, jammed ContScout.

This algorithm works on the proteins, like 2 other tools. The proteins are queried against a database associated to a taxonomy.

This produces hits that are further trimmed to produce High Level taxon (HLT) for each protein.

These HLT are then summarized at the contig level to filter or not all the proteins from the contigs.

This tool is tested on simulated data and on manually curated data.

The effect of genomic contamination on ancestral state reconstruction is also tested.

I found this article a little bit chimeric with two messages, one about a new algorithm of contamination detection and one about the effect of contamination detection on ancestral state reconstruction.

The first part is the most important section in terms of impact and the second one depends on this first section. My comments are thus mainly on the first section.

I have major concerns about the algorithm which lead to a major review of this paper.

I encourage the authors to make changes in their algorithm and in the manuscript.

MAJOR:

A lot of algorithms exist in the recent literature to detect contamination.

ContScout works at the level of the proteins, but the problem is the same as for the tools that work on DNA.

This is not an advantage of ContScout to accept proteins. For the user, it is more complicated (especially for eukaryota) to provide proteins and GFF compared to providing only the genome. In my opinion, ContScout is not taking an empty place in the landscape of algorithms. Other tools, such as GUNC/CheckM/EukCC, works also at the levels of proteins, protein prediction being part of these tools. This should be clearly mentioned in line 78-86.

The HLT (archaea, bacteria, plant, fungi, animal, other eukaryote) limits the detection to contamination from a distant taxonomic origin, which reduces the interest of ContScout. The possibility that contaminations below this high taxonomic level pass the detection of ContScout reduces the impact of the study, the use of an additional tool is still needed to not include contaminant in a study. The introduction and testing of other levels for HLT is needed here.

Currently, ContScout deletes all proteins from a contig when it is considered as a contaminant. I understand that this is better than fully discarding a genome (the major advantage of ContScout) when it is too highly contaminated but it would be useful to also add a non-greedy mode where only the proteins from the contaminant part of the contig are deleted. In the same idea, it would be useful to produce the genome (by masking) along with the deleted proteins.

Figure 1, Part 3, Second plot: The gene in red is a contaminant but the contig pass. It is mentioned in the manuscript that no thresholds are used, compared to other tools. But, accepting this case is actually a kind of threshold. What if the user doesn't want any contaminant proteins?

I still can't understand how the taxonomy of the whole genome/proteome is defined, by the user?

Line 171-172/ Line 215-216: How can you exclude that this is not linked to HGT ?

Line 395-397: This should be demonstrated. The difference between HGT and Contamination is a very important issue.

An additional experiment on HGT is needed to affirm this, especially with the big difference in terms of detection between ContScot and BASTA/Conterminator.

Line 411-414: I am sorry but I found that just too easy. Currently, ContScout can be used on proteins to detect distant contamination. Saying that it can be easily extended to a finer taxonomic level in a future release is too easy. It is linked to my commentary on HLT. Metagenomic produce frequently contaminated bacterial bins, which might be contaminated at a finer taxonomic level and it is especially where ContScout can be useful. This should be tested within this manuscript and not in a future release. I found the algorithm quite similar to GUNC and I recommend that you add a section with a comparison to GUNC on bacterial genomes, but at a finer taxonomic distance (at least bacterial phylum contaminated by another bacterial phylum).

Line 518-521: « no evidence of contamination », based on what?

EukCC paper (Saary et al) is cited in your study but nothing is mentioned about the performance of ContScout vs EukCC, which is one of the most important tools for eukaryotes.

In the Github, it is mentioned that a detailed user documentation is being created. This should have been made before the submission of the article so that reviewers can also evaluate this.

This study is not reproducible. I ask to the authors to provide a supplemental file with the command lines used, for contamination detection and ancestral state reconstruction.

MINOR.

Line 49: Not only draft genomes can be contaminated

Line 71-74: Add references

Line 81: It is well known, but a reference is needed

Line 86: Not only Basta and Conterminator can handle proteins, Physeter and Busco too.

Line 106: The levels should be defined here.
Line 227: Can you add computation time in the comparison
Line 232-234: Not clear. ContScout match exactly 1476 proteins, not more, not less?
Line 383: Should be proteomes and not genomes
Line 388: Can you explain more the results and include a comparison of the algorithms

Reviewer #3 (Remarks to the Author):

The manuscript 'Purging genomes of contamination eliminates systematic bias from evolutionary analyses of ancestral genomes' by Balint et al. describes a protein-based approach to detect rampant contamination remaining in public genomes. This cluttering of the databases can lead to further propagation of the error and as shown by the authors can have severe effects when performing ancestral genome reconstruction. I found the manuscript well-written, with well-thought-out experiments and the manuscript addresses a problem that undeniably needs to be addressed.

I do have some major concerns to be shared with the authors:

The first and foremost, is that I couldn't get the code running. I found the user manual on Github very limited as it doesn't explain how to run the code or the meaning of any of the arguments.

1) The docker image mentioned in the paper doesn't seem to be the latest version of the code. The github page refers to `docker pull h836472/contscout:latest`, while in the paper they point to `h836472/contscout_avx2`. The one mentioned in the paper doesn't seem to have the database downloading script present.

2) When downloading the swissprot database in diamond format the `db_inventory.txt` was created doesn't seem to abide to the expectations of the ContScout script:

```
"swissprot" "swissprot/8bd3cc66/swissprot-prot-metadata.json"
"6d11a7d60b9c06359e8f29912295d792" "8bd3cc66"
"swissprot/8bd3cc66/diamond/8bd3cc66_swissprot_tax.db" "8bd3cc66" "568796" "ccedf1c7"
"2023-02-22 09:18:49" "mmseq" "yes"
```

instead of:

```
"swissprot" "swissprot/8bd3cc66/swissprot-prot-metadata.json"
"6d11a7d60b9c06359e8f29912295d792" "8bd3cc66"
"swissprot/8bd3cc66/diamond/8bd3cc66_swissprot_tax.db.dmnd" "8bd3cc66"
"568796" "ccedf1c7" "2023-02-22 09:18:49" "diamond" "yes"
```

3) Once this was fixed, there is an error in the error message, it said `gff` files need to be stored in the folder `GTF_annot/` instead of `GFF_annot/`

4) Finally, when running Contscout as a singularity image the pipeline crashed after the diamond search. It wanted to use my local R installation of the package `stringi` and I believe it fails at line 476: `print_screen(hux(protium))`

```
Reported 302065 pairwise alignments, 302065 HSPs.
```

```
18550 queries aligned.
```

```
Now reading Alignment result (ABC) file.
```

```
Classifying individual proteins.
```

```
All_vote_same T/F:10075/7507.
```

Protein tag summary:

```
TaxTag Nprot
```

```
Error in dyn.load(file, DLLpath = DLLpath, ...) :
```

```
unable to load shared object '$localdir/R/x86_64-pc-linux-gnu-library/4.1/stringi/libs/stringi.so':  
libcui18n.so.60: cannot open shared object file: No such file or directory
```

```
Calls: print_screen ... ncharw -> loadNamespace -> library.dynam -> dyn.load
```

```
Execution halted
```

Moreover, I have some remaining questions and suggestions on the manuscript:

1) Please expand on the description of the methodology in the text please, as this is now very concise and is better explained in figure legend.

2) In the introduction, the benefit of protein-based methods is described compared to DNA-based ones. Could the analysis be expanded by including one such method to support these claims, e.g. perhaps FCS (<https://github.com/ncbi/fcs>) that is developed at NCBI to screen genomes prior to submission to Genbank.

3) I personally believe a more logical flow of the paper would be to first look at synthetic data, then the manually filtered genomes to validate your approach further and then go into the dataset of 844 eukaryotic genomes.

4) Although the G36 set is explained in the methods, briefly explain this set at L138. I find this sentence very confusing, why not say: Performance of ContScout was assessed by creating artificially contaminated genomes originating from 17 contamination-free genomes (reference to list). For all pairwise combinations of these 17 genomes that don't belong to the same HLT, 100, 200, 400, 800, 1600 or 3200 contaminant donor proteins were inserted in the receiver proteome. Please can it be confirmed that the proteomes of none of the 17 proteomes are part of the reference uniref100 database. Does the section contscout run parameters (line 510) refer to only the 844 run, or also the synthetic dataset.

5) Regarding the methods on L514, what do you mean with:

Contamination from all possible high level taxa (i.e. archaea, bacteria, plant, fungi, animal, other eukaryote, referred in the rest of the article as HLT) was screened (-x all).

6) You refer to Figure 3I,II (L202,L206,L210,L213) in the text while this is Figure 4.

7) Please explain taxon support value calculation both briefly in the results (L206-forward) and in the methods.

RESPONSE TO REVIEWERS' COMMENTS

Reviewer #1

Review of the manuscript „ContScout: a novel tool for accurate genome decontamination with critical impact on the analysis of ancestral genomes” submitted by Balint et al.

In this manuscript, the authors present a novel approach for sequence-similarity based contamination detection in gene sets inferred from genome (draft) sequences. Each protein sequence is searched against UniRef100, and a dynamic threshold is applied to assign the sequence to a higher order taxon, where the considered taxa are at least on the kingdom level. On this basis, they propose to decontaminate assemblies by removing contigs/scaffolds of which at least half of the genes have been flagged as contaminants. The authors further present several analyses to show the impact undetected contaminants can have on downstream evolutionary analyses including the assessment of ancestral gene set sizes as well as the prevalence of lineage-specific gene gains and losses. Overall, the analysis touches on a relevant topic that was addressed by several publications that appeared in the last years. The findings reported here reproduce that genome assemblies in the public domain are, in parts, heavily contaminated. They further support the general understanding that contaminations can be positively misleading in evolutionary analyses that are unaware of their existence.

Authors: We appreciate the Reviewer's comments and are happy to revise the ms in accordance with the suggestions given below. We note that our ms significantly improves on several aspects that the Reviewer mentioned. We demonstrate that ContScout outperforms existing tools for contamination detection, and, in the new version, even works at finer taxonomic scales. We think our results not only support the notion that contamination misleads evolutionary analyses, but also document the patterns of bias and mechanistically explain how contamination does so.

Major issues

1. Taxonomic assignment of protein sequences is well established in the literature, and it is common to apply a least common ancestor approach (LCA) on a dynamically selected set of Blast/Diamond top hits (see e.g. Huson et al. 2007; DOI: 10.1101/gr.5969107; and an LCA-based taxonomic assignment is also implemented into Diamond). The authors here propose their own rule-based approach considering a dynamically selected set of top hits resulting from a database search with either Diamond or MMSeq2. Since the approaches are conceptually similar--with the difference that the LCA approach integrates the taxonomic assignments of the considered hits instead of going for a majority vote--they should be exhaustively

compared. A specific focus should be placed here on the simulations where the rule-based approach presented here did not perform well, e.g. contaminations by *Dictyostelium*.

Authors: Our revised manuscript presents comparisons to three LCA-based approaches, BASTA, DIAMOND and MMSeqs. The results indicate that ContScout outperforms these approaches. We refer the Reviewer to the relevant new section in the manuscript (chapter: Comparison of ContScout with other tools for detecting contamination), here we only highlight the case of *Quercus suber*. Using ContScout, we discover a massive fungal contamination in *Q. suber*, which is confirmed by a BUSCO screen with Ascomycota specific HMM profiles. For an unbiased view, we use the BUSCO hits as taxon-specific markers and check their distributions among the assembled contigs. We selected 35 contigs, each having more than 100 proteins, out of which at least 20 were classified by BUSCO as fungal-specific while none of the proteins matched any plant specific profile. We checked the taxon tag distribution for the 7955 proteins encoded by these contigs and found that 86-96% of them are mislabelled by LCA methods (MMSeqs, Diamond), as shown by taxon tag statistics on 35 chosen contigs

ContScout protein tags:

fungi	7842(99%)
nodata	77
ambig	19
bacteria	15
metazoa	1
viridiplantae	1

MMSeqs protein tags:

- 6859 *Quercus suber* (86%)
- 1008 Eukaryota
- 69 Cellular organisms
- 6 Root
- 3 leotiomyceta
- 2 *Rhinocladiella mackenziei* CBS 650.93
- 2 Dothideomycetes
- 2 *Cercospora*
- 2 Fungi
- 1 *Mycosphaerellales*
- 1 *Rachicladosporium*

Diamond protein tags:

- 7608 *Quercus suber* (96%)
- 314 Eukaryota

27 Cellular organisms
4 root
2 Opisthokonta

We note that the LCA approach gives taxonomic tagging only, after which a separate rule set is needed to decide what should be removed as contamination. Therefore, comparability to MMSeqs/Diamond is limited to the taxonomic tagging step, not the contamination identification step.

The weaker performance of the previous ContScout version on synthetic data where *Dictyostelium* was the source of contamination can be attributed to the limited sampling density of the genus in the reference database and a strict hit filtering rule that was applied. In the previous version of the tool, at least three separate hits were required to classify a query protein. Nearly 50% of the query proteins from *Dictyostelium* did not meet this rule, resulting in frequent "no data" calls. These untagged proteins largely remained undetected during the simulation.

When modifying the module for labelling protein taxons in ContScout, we removed this filtering rule. This way, in the latest version of the tool, a single hit from the reference database is sufficient to label the query protein, albeit with a very low confidence value. We draw the Reviewers' attention to Figure 2, in which *Dictyostelium* - with the improved taxon tagging - was consistently detected as contaminant in the synthetic mixtures, even when testing at the finest taxonomic levels (i.e. family, order).

We would also like to draw the Reviewer's attention to the synthetic mixture between *Acanthamoeba castellani* and *Homo sapiens* (Supplementary Figure 4.). Even with the more permissive taxon labelling rule, more than 45% of query proteins at each taxon level are associated with "no_data" because no closely related sequence is present in the reference database. Accordingly, ContScout's recognition performance remains very limited for this particular synthetic mixture. (Median AOC: 0.61, IQR: 0.60-0.62)

To inform the user of cases that may limit detection performance, ContScout automatically provides a warning message if the proportion of unknown proteins in the query sequence is high. Similarly, the tool provides summary statistics on the number of independent hits supporting each taxon call at the protein level. This feedback can help the user decide whether ContScout analysis of a particular genome and database combination is feasible at a particular taxon level.

2. The contamination screen is limited in its resolution because sequences are assigned only to higher-order taxa on the level of kingdoms and beyond. This

precludes the identification of contaminations by sequences from the same kingdom or domain. Although the authors state that it is straightforward to increase the resolution of their taxonomic assignment, I consider it essential that this is shown. Essentially, the LCA approach does not share this limitation, and hence the authors would need to show that their rule-based approach implemented into ContScout extends the state-of-the-art in the field.

Authors: We have extended the algorithm to test for contamination at finer taxonomic scales and demonstrate in the revised ms that, ContScout can reliably separate contaminant and host proteins even at the family level. The new implementation tests for contamination at each NCBI taxonomic level (superkingdom, kingdom, phylum, class, order, family) and reports diagnostics that help the user make a call on the finest resolution where separation is feasible. This is necessary because the precision of host-contaminant separation depends on taxon sampling density and thus the resolution of detection may vary from taxon to taxon. After extensively testing the new algorithm, we find that high taxonomic resolution can be used in better-sampled clades (animals, fungi, plants), whereas in undersampled taxa (e.g. unicellular eukaryotes) calls even on the level of kingdoms can be problematic. We expect this will improve as more and more genomes become available

We demonstrate the performance of the updated algorithm using synthetic data tested across a range of NCBI taxonomic levels and case studies and synthetic datasets comprising mixtures of genes from the mosquito and malaria parasites (Plasmodium), another insect genome and a parasitic wasp, two bacteria bacterium, or the mouse genome contaminated with human sequences, among others. We also created a pair of Candida and Saccharomyces, two yeast species from the same family. These results were included in the chapter "Performance assessment on synthetic data" and Figure XX.

3. The authors describe the impact that contaminations have on the outcome of evolutionary studies, where they basically result in the overestimation of gene age, gene loss events, and of HGT (which is not considered here). This topic has received considerable attention over the past years (see e.g., Cornet and Baurain 2022). Although it is nice to see numbers on a tree (see Figs. 5 and 6), the corresponding values heavily depend on the precise taxa under study, and with the granularity of the contaminant detection (see above). Thus, what is presented here is hard to generalize, with the exception of the amounts of foreign genes in the individual assemblies. It is necessary to carve the novelty in the present study.

Authors: We respectfully disagree with this comment. The impact of contamination on evolutionary studies has indeed received attention recently, however, to the best of our knowledge, no empirical measurement of its impact is available, nor has it been clear how contamination causes gene origins to be pushed further back in time revealed.

We agree with the reviewer that the numerical values depend on the taxa included in the dataset, which is a property of all evolutionary studies of gene content. Accordingly, we did not use these estimates to explore deeply the biology of ancestral eukaryotic taxa. However, our analyses outline clear trends that stem from contamination, which we think are generalizable. First, ancestral gene content is overestimated if contamination is present in the data. Second, contamination from multiple taxa additively inflates ancestral gene content estimates. We show how these effects can affect deep eukaryote nodes and ones that connect bacteria with eukaryotes. Overall, we shortened this section to give more space to new analyses related to contamination detection, nevertheless, we feel this section has novelty and provides a mechanistic explanation for a phenomenon that has been discussed in the literature several times.

4. How were the 36 taxa selected that formed the data basis for studying the evolutionary implications of contaminations? It is surprising that several taxa, especially within the animals, are highly overrepresented: *Caenorhabditis* – 2 species, *Bombus* – 2 species, *Drosophila* 3 species.

Authors: This dataset was assembled so that it (1) is of modest size that allows computational tractability even with several versions of it being analyzed (2) includes all major eukaryotic groups and (3) includes close non-contaminated relatives of contaminated taxa. The genera mentioned by the Reviewer are represented by more than one species because one of these is contaminated and the other(s) not, so that the non-contaminated species can serve as a control (induces a minimum number of duplications implied by the presence of that genus).

5. On page 11, the authors investigate why contaminations inflate gene loss estimates nearly five times more than gene gain estimates. This finding is conceptually easily interpretable, because a contamination (again similar to a horizontal gene transfer) introduces a foreign gene into one taxon but not into its relatives. If we use a naïve Dollo parsimony-based approach of reconstructing the evolutionary fate of the gene, this will lead to a vast overestimation of the gene age resulting in an inflated number of only gene losses but rarely of gene gains. If the authors plan to keep this analysis in the results, it would be very helpful if they could provide additional motivation on why they think that this finding is important to report.

Authors: We agree that these mechanisms are relatively easy to interpret for an evolutionary biologist. However, for non-evolutionists, who we expect will dominate the readership of this paper, demonstrating how contamination inflates gene loss estimates might be important. Further, while speculations and theoretical considerations exist in the literature for this, to the best of our knowledge no actual

measurements have been published. Nevertheless, we shortened this part of the manuscript, to put more emphasis on the decontamination algorithm.

6. The tree, as it is shown in Figs 5 and 6 reflects the Coelomata-hypothesis (Drosophila and human grouped to the exclusion of *C. elegans* instead of the commonly accepted Ecdysozoa-hypothesis (Drosophila and *C. elegans* grouped to the exclusion of humans). If the authors want to keep this phylogeny, they should explain why this is justified.

Authors: We appreciate this comment and re-inferred the phylogeny using a constraint that reflects the Ecdysozoa hypothesis (the maximum likelihood solution for our dataset is the Coelomata hypothesis, which is not surprising given the small dataset) and re-run the relevant analyses.

7. The definition of orthoGroups in the text (line 355-356) reads like 1:1 orthologs where the orthoGroup represents a clique of orthologous proteins (the orthology assignment criterion is fulfilled for each pair of proteins in the orthoGroup, see e.g. OMA groups). This would be a very strict criterion resulting in many but typically very small orthoGroups. More inclusive approaches integrate also inparalogs into the orthologous groups (see OMA hierarchical orthologous groups). The authors should increase the level of precision here. Moreover, the context in which the orthologous groups are mentioned is not clear. The inclusion of a contamination into an orthologous group will always inflate a gene age estimate by Dollo parsimony irrespective of how the group was compiled. In fact, I would expect that more inclusive orthology assignments likely suffer from a stronger impact of contaminations because of a tendency to generate larger orthologous groups. This is contrary to what is stated in the manuscript where the authors mention their expectation that 'simpler' approaches are less sensitive to the distorting effects of contamination. In any case, such speculations should be replaced by an analysis that is backed up by data.

Authors: We shortened this section considerably and removed the discussion on simpler orthogroups.

Minor issues

1. What is RLE?

Authors: we removed this technical term (referring to Run Length Encoding) from the ms.

2. The authors state that people use fixed-size hits for taxonomic assignments (e.g. Top100). This is not state of the art (see MEGAN), and should be presented in a more differentiated manner.

Authors: We rephrased the quoted section.

3. Caption of Fig. 3 could be condensed

Authors: Done.

4. The heading 'Phylogenomic analyses are biased by contamination' should be rephrased, since many people connect with 'Phylogenomics' the inference of evolutionary relationships between species using large taxon-gene matrices. This, however, is not in the focus here.

Authors: Done.

5. What is the rationale of limiting the computation of the taxonomy support value (Fig. 4) to the top ten hits when the taxonomic assignment uses a dynamic approach?

Authors: This is meant as an independent measure of taxonomy support. In establishing taxonomy support for this section (performance comparison of different tools), we intentionally chose a method that does not mirror ContScout's internal logic, in order to disfavour ContScout in these comparisons.

6. In the discussion, the authors mention the application of ContScout in the context of decontamination of genome assemblies by removing human sequences. Removal of contaminating human sequences is routinely done on the read level prior to genome assembly, and rapid kmer-based approaches, such as Kraken2 are perfectly suited for this application since the contaminating species is known, and a finished genome assembly is at hand.

Authors: We removed this statement from the ms??.

7. HLT should be explained when it occurs first in the text, since this clarifies right from the start the limits in resolution.

Authors: Because the resolution of ContScout has been extended, we removed the HLT approach from the paper.

Reviewer #2

Bálint et al., present a new algorithm for contamination detection, jammed ContScout.

This algorithm works on the proteins, like 2 other tools. The proteins are queried against a database associated to a taxonomy. This produces hits that are further trimmed to produce High Level taxon (HLT) for each protein. These HLT are then summarized at the contig level to filter or not all the proteins from the contigs. This tool is tested on simulated data and on manually curated data. The effect of genomic contamination on ancestral state reconstruction is also tested.

I found this article a little bit chimeric with two messages, one about a new algorithm of contamination detection and one about the effect of contamination detection on ancestral state reconstruction.

The first part is the most important section in terms of impact and the second one depends on this first section. My comments are thus mainly on the first section. I have major concerns about the algorithm which lead to a major review of this paper.

I encourage the authors to make changes in their algorithm and in the manuscript.

Authors: We appreciate the Reviewers' comments and reworked the ms in accordance with the suggestions.

MAJOR:

1., A lot of algorithms exist in the recent literature to detect contamination. ContScout works at the level of the proteins, but the problem is the same as for the tools that work on DNA.

This is not an advantage of ContScout to accept proteins. For the user, it is more complicated (especially for eukaryota) to provide proteins and GFF compared to providing only the genome.

In my opinion, ContScout is not taking an empty place in the landscape of algorithms. Other tools, such as GUNC/CheckM/EUkCC, works also at the levels of proteins, protein prediction being part of these tools. This should be clearly mentioned in line 78-86.

Authors: We agree that the field has been very active recently, with multiple tools available for analysing protein/DNA data. Thus, though ContScout may not take a fully empty space, it improves tremendously over other (LCA-based) tools and has additional features, which we think we demonstrated and now even better documented in the revised manuscript. Most importantly, tools available currently can only assign taxonomic tags to genes/proteins, whereas ContScout has, in addition, a rule-based approach for deciding which proteins to remove from the genome as contamination. Thus it does not only detect the presence of

contamination, but pinpoints exact proteins that are a result of that. We are confident that ContScout will have a broad user base, in particular considering the recent widening of genome sequencing efforts in hitherto lesser sequenced taxa. We also envision that ConstScout may be picked up by the metagenomics community, where currently LCA-based approaches dominate, of which the tools tested by us showed inferior performance to ContScout.

2., The HLT (archaea, bacteria, plant, fungi, animal, other eukaryote) limits the detection to contamination from a distant taxonomic origin, which reduces the interest of ContScout. The possibility that contaminations below this high taxonomic level pass the detection of ContScout reduces the impact of the study, the use of an additional tool is still needed to not include contaminant in a study. The introduction and testing of other levels for HLT is needed here.

Authors: We appreciate this comment, and extended the algorithm to test for contamination at finer taxonomic levels. The new implementation tests for contamination at multiple taxonomic levels simultaneously, and provides diagnostics for finding the lowest taxonomic category at which taxon sampling density allows highly specific detection (see also our answer to Reviewer 1.).

3., Currently, ContScout deletes all proteins from a contig when it is considered as a contaminant. I understand that this is better than fully discarding a genome (the major advantage of ContScout) when it is too highly contaminated but it would be useful to also add a non-greedy mode where only the proteins from the contaminant part of the contig are deleted. In the same idea, it would be useful to produce the genome (by masking) along with the deleted proteins.

Authors: This is a good idea, and we agree deleting only contiguous genes flagged as contamination will be a useful feature. However, we decided against implementing it in this revision, for two reasons. First, time constraints did not allow us to experiment with a new non-greedy algorithm and second (more importantly), a non-greedy version might also remove horizontally transferred genes which is currently a non-preferred behaviour and would require a lot more research. We note that the easy-to-implement intermediate solution (delete only proteins, irrespective of scaffold membership) performed poorly in our benchmarks.

4., Figure 1, Part 3, Second plot: The gene in red is a contaminant but the contig pass. It is mentioned in the manuscript that no thresholds are used, compared to other tools. But, accepting this case is actually a kind of threshold. What if the user doesn't want any contaminant proteins?

Authors: The Reviewer misunderstood this figure (due to insufficient explanation of coloring): the gene in red is not directly called as a contaminant, what we conclude at stage II. is that hit lists do not support its taxonomic affiliation. In Stage III, this protein is saved due to a majority rule voting across the scaffold. Proteins like this can derive from HGT, which ContScout algorithm intentionally does not discard (in the revision this is demonstrated in a new chapter).

The Reviewer is right, that the majority voting is indeed an implicit threshold. However, we do not claim in the ms that ContScout is completely threshold-free, what we emphasised is the lack of that at the sequence search stage.

5., I still can't understand how the taxonomy of the whole genome/proteome is defined, by the user?

Authors: Query genome taxonomy information is set by the user via the “-q” switch when starting the ContScout analysis. This parameter expects a single NCBI taxon ID that is associated with the organism whose genomic data is to be screened for contamination. The user-provided taxonID is automatically translated to expected superkingdom, kingdom, phylum, class, order and family values based on the taxonomy database, automatically imported from NCBI. This reference taxon data is later used by the tool to decide which contig to accept as host and which to tag as potential contamination.

6., Line 171-172/ Line 215-216: How can you exclude that this is not linked to HGT ?
Line 395-397: This should be demonstrated. The difference between HGT and Contamination is a very important issue.

An additional experiment on HGT is needed to affirm this, especially with the big difference in terms of detection between ContScot and BASTA/Conterminator.

Authors: We provide new analyses, which demonstrate that ContScout can efficiently discriminate HGT from contamination.

7., Line 411-414: I am sorry but I found that just too easy. Currently, ContScout can be used on proteins to detect distant contamination. Saying that it can be easily extended to a finer taxonomic level in a future release is too easy. It is linked to my commentary on HLT. Metagenomic produce frequently contaminated bacterial bins, which might be contaminated at a finer taxonomic level and it is especially where ContScout can be useful. This should be tested within this manuscript and not in a future release. I found the algorithm quite similar to GUNC and I recommend that you add a section with a comparison to GUNC on bacterial genomes, but at a finer taxonomic distance (at least bacterial phylum contaminated by another bacterial phylum).

Authors: Following the advice, we updated the ContScout algorithm and tested it on artificially contaminated genomes where we mixed genomes of taxa belonging to the same phylum, same class, same order or family. ContScout performed very well in these tests, though it should be noted that performance at finer taxonomic levels (order, family) depend on taxon sampling density. The results have been incorporated in the chapter “Performance assessment on synthetic data”.

We also carried out a comparison between ContScout and GUNC. Please note that GUNC does not work with eukaryotic sequences (limited by its reference database and the prokaryote-specific built-in gene finder tool, Prodigal), therefore, we could not directly benchmark it the same way we tested the sensitivity of ConTerminator, BASTA, MMSeqs and Diamond. Furthermore, differences between GUNC and ContScout make it generally very challenging to perform an accurate and comprehensive comparison.

We found that both tools detect the presence of the massive contamination starting from the “order” level, where the taxonomic lineage of the two organisms first separate. The 0.57:0,43 Pseudomonas:Escherichia mixing ratio is perfectly captured by both tested tools.

Additional details on analytical steps taken to make the two software comparable are provided below:

*1., GUNC expects **DNA sequences** and carries out protein annotation on its own (based on prokaryotic gene finder tool Prodigal). ContScout expects **protein sequences** together with a **GFF/GTF protein annotation** file.*

Actions taken to allow comparison:

We carried out a GUNC analysis on a synthetic bacteria-bacteria mix where contig DNA sequences of Escherichia coli (GCF_000731455.1) - Pseudomonas aeruginosa (GCF_000710625.1) were pooled. GUNC was called via the command

```
“gunc run -i Escholi_Pseuaeru_mix.fna -r  
/work/balintb/databases/gunc/gunc_db_progenomes2.1.dmnd --detailed_output  
--contig_taxonomy_output --temp_dir /work/tmp -o Eco_Pseu_mix_GUNC”
```

Later, the predicted protein sequence generated by GUNC/Prodigal together with a hand-made GFF annotation file based on the contig information found in the protein header was used to perform the ContScout run on the synthetic data.

2., Available reference database of GUNC and available reference databases for ContScout largely differ.

Actions taken to allow comparison: the installed GUNC reference database was manually converted to a database format that ContScout could use during the analysis.

3., While GUNC performs a thorough quality control assessment, and reports the presence or absence of contamination at various taxon ranks (kingdom, phylum, class, order, family, genus, species), it does not itself classify proteins or contigs as “host” or “contaminants”. Even when executed with the “--contig_taxonomy_output” parameter, it fails to provide a single taxon label at any taxon rank for any of the contigs. Instead, it gives a summary about the number of proteins supporting each observed taxon label per contig.

Actions taken to allow comparison: We applied the same consensus contig calling rule on the GUNC contig taxon call data, as the one ContScout uses.

*4., Despite providing contamination summary on all levels, Not all taxon level appears in the contig taxonomy output of GUNC. That way, we were limited to carrying out the comparison between the two tools on the **family** level.*

Even with all efforts made, we feel that the comparison between GUNC and ContScout is not solid enough to be included in the main manuscript. Instead, we provide the analysis results as part of the response to Referees.

8., Line 518-521: « no evidence of contamination », based on what?

Authors: explanation added.

9., EukCC paper (Saary et al) is cited in your study but nothing is mentioned about the performance of ContScout vs EukCC, which is one of the most important tools for eukaryotes.

Authors: EukCC is a QC tool that can detect the presence of contamination, but cannot identify contaminating proteins. Therefore, we think the two tools are not comparable, with ContScout doing a more thorough job, by identifying specific contaminating proteins. EukCC uses single copy gene sets, whereas ContScout tests each protein in the genome.

10., In the Github, it is mentioned that a detailed user documentation is being created. This should have been made before the submission of the article so that reviewers can also evaluate this.

Authors: We apologise for this delay. The User documentation for the improved code is now available on Github.

11., This study is not reproducible. I ask to the authors to provide a supplemental file with the command lines used, for contamination detection and ancestral state reconstruction.

Authors: Source data including scripts and command line parameters have been collected and shared at figshare.

MINOR.

Line 49: Not only draft genomes can be contaminated

Authors: Corrected.

Line 71-74: Add references

Authors: references are provided a few lines below, where we mention specific software for each approach.

Line 81: It is well known, but a reference is needed

Authors: We think the Reviewer refers to this statement here: "Because DNA evolves faster than protein sequence". We added a reference.

Line 86: Not only Basta and Conterminator can handle proteins, Physeter and Busco too.

Authors: We added a reference to Physeter, whereas for BUSCO we note its more of a QC tool than a contamination detector.

Line 106: The levels should be defined here

Authors: This part was rewritten to reflect the new ContScout algorithm

Line 227: Can you add computation time in the comparison

Authors: BASTA search for the 200 genomes took place in a single batch run at the NERSC supercomputing facility using a python-based MPI massively parallel task processing framework. The runtime for individual BASTA tasks were not recorded. We currently have no access to the cluster usage accounting system where the global run parameters (number of nodes booked, project total runtime) could be extracted. Thus, we currently can not provide a comprehensive runtime estimate for all compared tools.

Line 232-234: Not clear. ContScout match exactly 1476 proteins, not more, not less?

Authors: Yes, that is true. We rephrased this section slightly.

Line 383: Should be proteomes and not genomes

Authors: Corrected.

Line 388: Can you explain more the results and include a comparison of the algorithms

Authors: We added explanations.

Reviewer #3 (Remarks to the Author):

The manuscript 'Purging genomes of contamination eliminates systematic bias from evolutionary analyses of ancestral genomes' by Balint et al. describes a protein-based approach to detect rampant contamination remaining in public genomes. This cluttering of the databases can lead to further propagation of the error and as shown by the authors can have severe effects when performing ancestral genome reconstruction. I found the manuscript well-written, with well-thought-out experiments and the manuscript addresses a problem that undeniably needs to be addressed.

Authors: We appreciate the Reviewer's suggestions and have incorporated them in the ms. We double checked that the new version runs smoothly, as described in the updated documentation.

I do have some major concerns to be shared with the authors:

The first and foremost, is that I couldn't get the code running. I found the user manual on Github very limited as it doesn't explain how to run the code or the meaning of any of the arguments.

1) The docker image mentioned in the paper doesn't seem to be the latest version of the code. The github page refers to *docker pull h836472/contscout:latest*, while in the paper they point to *h836472/contscout_avx2*. The one mentioned in the paper doesn't seem to have the database downloading script present.

Authors: We apologise for the inconvenience. Unfortunately, version conflicts likely did happen.

*To make the manuscript version vs code version assignments unambiguous, we created a frozen branch in GitHub named **bioRxiv_version** corresponding to the outdated code version referred by the bioRxiv manuscript version*

(<https://doi.org/10.1101/2022.11.17.516887>). Matching binary can be downloaded from Docker via `docker pull h836472/contscout:biorexiv`

The improved code documentation that is linked to the present revised manuscript can be found frozen under the branch name **NatComm** with a matching binary available under DockerHub via `docker pull h836472/contscout:natcomm`

Source code for any further version along with the latest bug fixes shall appear under the **main** branch in GitHub with matched binary placed under `docker pull h836472/contscout:latest`.

2)When downloading the swissprot database in diamond format the `db_inventory.txt` was created doesn't seem to abide to the expectations of the ContScout script:

```
"swissprot" "swissprot/8bd3cc66/swissprot-prot-metadata.json"  
"6d11a7d60b9c06359e8f29912295d792" "8bd3cc66"  
"swissprot/8bd3cc66/diamond/8bd3cc66_swissprot_tax.db" "8bd3cc66" "568796"  
"ccedf1c7" "2023-02-22 09:18:49" "mmseq" "yes"
```

instead of:

```
"swissprot" "swissprot/8bd3cc66/swissprot-prot-metadata.json"  
"6d11a7d60b9c06359e8f29912295d792" "8bd3cc66"  
"swissprot/8bd3cc66/diamond/8bd3cc66_swissprot_tax.db.dmnd" "8bd3cc66"  
"568796" "ccedf1c7" "2023-02-22 09:18:49" "diamond" "yes"
```

Authors: Once again, we apologise for the inconvenience caused by the bug in the database updater module. In the improved version, the entire local database storage structure has been re-designed from scratch simplifying database registration and handling.

3)Once this was fixed, there is an error in the error message, it said `gff` files need to be stored in the folder `GTF_annot/` instead of `GFF_annot/`

Authors: We are grateful for the feedback on inconsistent input folder names (`GFF_annot` / `GTF_annot`). In the improved version, input folder nomenclature has been re-designed with "`protein_seq`" folder holding the protein sequence and "`annotation_data`" holding the genome annotation regardless to the annotation file format (`GFF`, `GTF`, `GFF3`)

4)Finally, when running Contscout as a singularity image the pipeline crashed after the diamond search. It wanted to use my local R installation of the package `stringi` and I believe it fails at line 476: `print_screen(hux(protium))`

*Reported 302065 pairwise alignments, 302065 HSPs.
18550 queries aligned.
Now reading Alignment result (ABC) file.*

Classifying individual proteins.
All_vote_same T/F:10075/7507.

Protein tag summary:

TaxTag Nprot

Error in dyn.load(file, DLLpath = DLLpath, ...) :

unable to load shared object '\$localdir/R/x86_64-pc-linux-gnu-library/4.1/stringi/libs/stringi.so':

libcui18n.so.60: cannot open shared object file: No such file or directory

Calls: print_screen ... ncharw -> loadNamespace -> library.dynam -> dyn.load

Execution halted

Authors: We think the problem here was, at least in part, caused by conflicts between local and containerized R installations. We respectfully advise against mixing host-based R installation with a containerized system. The image provided has been designed to contain all requirements for the tool (MMSeqs, Diamond, Jacksum, R together with all required libraries) and should work seamlessly on their own without any installation or re-configuration being needed from the user's side.

If tool customization is needed, the composition of the image can be conveniently altered by modifying the Docker build script followed by a local compile step. (Please note that the time needed for a build can be 40-80 minutes)

If desired, the ContScout R script together with the UpdateDB module can be downloaded and directly installed on the host server, thus integrating it into a local R environment. Instructions on a direct install are provided at GitHub.

Moreover, I have some remaining questions and suggestions on the manuscript:

1) Please expand on the description of the methodology in the text please, as this is now very concise and is better explained in figure legend.

Authors: We expanded the description of the algorithm in the Results section.

2) In the introduction, the benefit of protein-based methods is described compared to DNA-based ones. Could the analysis be expanded by including one such method to support these claims, e.g. perhaps FCS (<https://github.com/ncbi/fcs>) that is developed at NCBI to screen genomes prior to submission to Genbank.

Authors: We appreciate this comment, but think that comparison to FCS isn't necessary, because of the large number of differences between the two software and DNA vs protein approaches. The FCS-GX module, which aims to find sequences from other organisms, assumes that the genome of the contaminating

organism is already in an NCBI database. This is most often not the case, and hence we consider FCS as a tool to primarily remove human and vector/adaptor sequences.

3) I personally believe a more logical flow of the paper would be to first look at synthetic data, then the manually filtered genomes to validate your approach further and then go into the dataset of 844 eukaryotic genomes.

Authors: We restructured the ms as suggested.

4) Although the G36 set is explained in the methods, briefly explain this set at L138. I find this sentence very confusing, why not say: Performance of ContScout was assessed by creating artificially contaminated genomes originating from 17 contamination-free genomes (reference to list). For all pairwise combinations of these 17 genomes that don't belong to the same HLT, 100, 200, 400, 800, 1600 or 3200 contaminant donor proteins were inserted in the receiver proteome.

Authors: the section covering the performance on synthetic data has been completely re-written to accommodate closely related source-recipient pairs, separated at fine taxon levels. Reference to the genomes used as source or recipient in the synthetic contamination studies have been provided (Supplementary table 1)

Please can it be confirmed that the proteomes of none of the 17 proteomes are part of the reference uniref100 database. Does the section contscout run parameters (line 510) refer to only the 844 run, or also the synthetic dataset.

Authors: Although genomes used in the simulation are likely present in reference databases (UniRef100), ContScout has a specific filter that removes all hits that share the same taxonID as the query sequence. This filter effectively eliminates the undesired case when a query sequence could be confirmed by its exact copy from the public database. We believe that the use of this filter allows ContScout to outperform LCA-based methods such as MMSeqs and Diamond. (See Figure 4.)

5) Regarding the methods on L514, what do you mean with: Contamination from all possible high level taxa (i.e. archaea, bacteria, plant, fungi, animal, other eukaryote, referred in the rest of the article as HLT) was screened (-x all).

Authors: the -x parameter allows selecting certain groups as sources of contamination.

6) You refer to Figure 3I,II (L202,L206,L210,L213) in the text while this is Figure 4.

Authors: Corrected.

7) Please explain taxon support value calculation both briefly in the results (L206-forward) and in the methods.

Authors: Explanation of taxon support value calculation was added to the MS both in the Methods and in the Results section.

REVIEWER COMMENTS

Reviewer #1 (Remarks to the Author):

In the revised version of this manuscript, the authors have addressed a substantial fraction of my initial concerns. I appreciate the comparative analysis to other tools that was also requested by a second reviewer, and which helps to now better embed ContScout into the state of the art in this field. However, after going through the novel version of this manuscript, I still encountered several issues that require additional attention. I hope that my comments contribute to further improvement.

Major issue

1. Page 2, line 62 – The statement “(...) whether and how contaminations affects their performance” does not reflect the state-of-the-art in the field for the following reasons: It is known that contamination can be mis-interpreted as horizontal gene transfer (see refs in the manuscript). It is also known that HGT results in an overestimation of gene age and distribution (see, e.g., Step 2 in <https://www.ncbi.nlm.nih.gov/pmc/articles/PMC3812327/>). From transitivity follows that contamination results in an overestimation of gene age and distribution. It is further known that orthology assignments form the basis of many evolutionary hypotheses, see Fig. 1 in <https://academic.oup.com/mbe/article/36/10/2157/5523206> and that of course orthology inference methods will also pick up contaminants. In the light of the above, it would be more appropriate to state the following: ‘from theory’ the effects of contamination on (i) the composition of orthologous groups/phylogenetic profiles, (ii) on the taxonomic distribution of the gene, and (iii) on inferences about the evolutionary age of a gene and of its evolutionary fate are crystal clear and predictable. But what is lacking is a quantification.
2. Page 6, lines 198-199 - I am puzzled by the statement in the text that 33% of the conterminator hits (I assume that the authors here actually refer to the assignment as contamination) have “(...) taxon support values above 0.75 indicating possible false positives among those hits.” (line 199). How does this reconcile with Fig. 3a where the Venn diagram shows that all but 128 assignments are shared with ContScout. Is ContScout the also wrong? Note, the same is true for Basta. It might be that I am overlooking something here, but this needs to be clarified. In essence, both alternative tools predict a subset of the ContScout contaminations. What is then the relevance of referring to ‘possible false positives’?
3. Along the same lines, how many proteins are flagged as contaminants without further evidence besides its placement together with proteins that were identified as contaminants using sequence similarity to database entires (impact of consensus call).
4. Line 215 “Sensitivity assessment on manually filtered genomes”. The analyses attest ContScout an excellent sensitivity that exceeds that of Conterminator and the other tools by far. What it missing here, however, is the specificity assignment. This would help the readers a lot to gain confidence in this novel method.

Minor issues

1. Page 2, line 64
 - a. please give a reference for a database-free taxonomic assignment tool
 - b. Placing Blast searches and k-mers next to each other connected by an ‘or’ suggests that they are two alternative approaches, which they are not. Blast is the name of a database search heuristic--which in fact also makes use of k-mers to speed up the search for promising hit sequences—whereas pure k-mer-based approaches are just relying on exact pattern matching.
2. Page 2, line 77 – I do not see the relevance of mentioning the phylogenomics studies here. The focus is on decontamination. What kind of downstream analysis a user has in mind, and whether this is done with a nucleotide- or amino acid alphabet, should be irrelevant here.
3. Page 2, lines 80-82. From what was written before, it becomes not clear why the authors conclude that efficient and sensitive tools are ‘currently lacking’. The results of the present study should not serve as a justification for this statement in the introduction.
4. Page 3, line 83 – I suggest removing the word ‘accurate’. This is a relative term and whether something is considered accurate depends on an ad hoc threshold.
5. Caption Figure 1

- a. taxonomy should be rephrased to taxonomic assignment. A query genome has no 'taxonomy' (<https://en.wikipedia.org/wiki/Taxonomy>).
- b. Case q in 1B is not 'all green' unless it is explained before what the meaning of the box is
- c. The y axis of the bar charts should be explained
6. Page 4, lines 141-142 – I think that the term 'biologically realistic contamination scenarios' is not really helpful. In essence, contamination is a lab-issue and has little to do with biology.
7. Page 4, Line 147 – The reference to Fig. 1 is wrong here

Minor discretionary Revisions

1. Fig. 2 – I am not happy with the term 'host species'. It is tempting to interpret this in a biological way, but this would be inappropriate.

Reviewer #2 (Remarks to the Author):

I would like to express my gratitude to the authors for incorporating the modifications into the manuscript. The section on finer taxonomic affiliations is satisfactory. However, I still have serious reservations about this study, particularly regarding the distinction between HGT and contamination. The authors demonstrate that ContScout does not confuse HGT and contamination at very broad taxonomic levels, especially in bacteria. However, it is precisely in bacteria where HGT is most frequent, and this is even more true at lower taxonomic levels. Relying on the new analyses to claim that ContScout distinguishes HGT and contamination is insufficient.

It was also clear in my review that, from my perspective, ContScout needed improvement in its algorithm to produce genomes as well, in order to meet the needs of the field. Eliminating contaminated proteins without addressing genomics is insufficient. Masking the corresponding part of the genome is an absolutely necessary task in my opinion for a publication. I also requested the possibility for the authors not to eliminate the entire contig but only the relevant protein(s), as I am concerned about the loss of important data. I did not request this as the default mode but as an option for the user. The response to this comment, "performed poorly in our benchmarks," is insufficient. I would have appreciated seeing these tests or even better, this option.

Furthermore, it seems straightforward to infer a contamination rate based on the results of ContScout and compare it to EukCC. I understand that EukCC operates differently, but a comparison would have been interesting. Dismissing my comment based on the fact that EukCC is a QC tool is not a sufficient response for me.

In summary, I do not recommend this paper for publication because my comments have not been adequately addressed. Modifying the algorithm to mask genomes, testing a non-greedy mode, comparing it to EukCC (which is used by many researchers working on fungi), and further analyzing HGT in bacteria vs bacteria are necessary tasks, in my opinion, for publication.

Finally, I want to point out to the authors that for my second point, they partially referred me to the response to reviewer 1 without specifying which response. I assume it is referring to point 2. In this response, the authors refer to Figure XX, which is not very informative. In general, the response to the reviewers was difficult to follow.

I would like to conclude by extending my utmost respect to the authors and expressing my gratitude for their efforts. I want to emphasize that throughout this process, I have maintained a professional approach and strived to be meticulous.

Reviewer #3 (Remarks to the Author):

I would like to congratulate the authors for their thorough work on the resubmission and I feel the manuscript has greatly improved. Moreover, thank you for the updated code base and the improved user manual on the Github page. I am happy to report a successful installation and run

of the algorithm.

However, I do have a few remaining concerns:

1) On line 374-377 the authors suggest the usage of ContScout as a routine screening method after genome assembly. However as pointed out by reviewer 2 as this a protein-based tool this can only happen after gene annotation, the very last (and often laborious) step before submission. I would therefore frame this more as a useful tool when embarking on large-scale comparative genomics projects, such as the ancestral gene content analysis in the paper. The benefit of using a protein methodology is the slower mutation rate of proteins vs DNA (line 73-75). However, to the best of my knowledge, I don't think the detection limits of nucleotide / protein divergence have been compared in the framework of contamination screening, especially relative to latest innovations in k-mer methods (such as FCS, which allows variation in the third-base position, see <https://www.biorxiv.org/content/10.1101/2023.06.02.543519v1.full>). The response of the authors on my second comment that these tools expect the genome of the contaminant to be present in the database, is wrong. While I theoretically acknowledge the benefit of proteins, it would be nice to see this compared as ContScout also lists the scaffolds to be removed (could be a supplementary figure).

2) The analysis of ancestral gene content is heavily dependent on the topology. I didn't notice this last time, but the sister relationship between Amoebozoa and Discoba is unexpected (e.g. Burki 2020). Could the phylogeny be adapted similar as done in R1 comment 6?

3) On line 437-438 you acknowledge that not only contamination but also incorrect gene annotation and HGT can impact ancestral gene reconstruction analyses. Next, you state that the latter two processes will mainly affect terminal branches and therefore have only a limited effect on gene losses. However I don't believe HGT, especially cross-kingdom HGT, mainly affect the distal portion of the tree. In a parsimonious method, one gain and several losses will be preferred over two independent gains. Could you please explain how does the effect of HGT compare to contamination on gene loss inflation levels?

Minor / Textual comments:

Line 43: add comma after fungi

Line 49: 'while sample mishandling and incorrect data processing can cause technical problems'. You have listed here a set of problems that can result in contamination. Sentence doesn't have a logical causal relationship.

Line 73: Please add closing parenthesis

Line 100: add the word 'or'

Line 252: scrutinizing instead of scrutinize. Please split this sentence in two to help readability.

Line 303: Please add closing parenthesis

On line 226 you mention 680 contaminant proteins in the *B. impatiens* genome, while on line 316 you mention 965. Is due to the existence of two different contaminated genome versions?

Line 345: add space between as and bacterial.

Line 439: Please split the sentence after the reference.

Section sensitivity assessment on manually filtered genomes:

Could you please mention how many (if any) genes were flagged by any of the tools that weren't manually removed in the datasets.

RESPONSE TO REVIEWERS' COMMENTS

Reviewer #1 (Remarks to the Author):

In the revised version of this manuscript, the authors have addressed a substantial fraction of my initial concerns. I appreciate the comparative analysis to other tools that was also requested by a second reviewer, and which helps to now better embed ContScout into the state of the art in this field. However, after going through the novel version of this manuscript, I still encountered several issues that require additional attention. I hope that my comments contribute to further improvement.

Answer: We appreciate the Reviewer's suggestions, and have reworked the ms in accordance with the guidelines given.

Major issue

1. Page 2, line 62 – The statement “(...) whether and how contaminations affects their performance” does not reflect the state-of-the-art in the field for the following reasons: It is known that contamination can be mis-interpreted as horizontal gene transfer (see refs in the manuscript). It is also known that HGT results in an overestimation of gene age and distribution (see, e.g., Step 2 in <https://www.ncbi.nlm.nih.gov/pmc/articles/PMC3812327/>). From transitivity follows that contamination results in an overestimation of gene age and distribution. It is further known that orthology assignments form the basis of many evolutionary hypotheses, see Fig. 1 in <https://academic.oup.com/mbe/article/36/10/2157/5523206> and that of course orthology inference methods will also pick up contaminants. In the light of the above, it would be more appropriate to state the following: ‘from theory’ the effects of contamination on (i) the composition of orthologous groups/phylogenetic profiles, (ii) on the taxonomic distribution of the gene, and (iii) on inferences about the evolutionary age of a gene and of its evolutionary fate are crystal clear and predictable. But what is lacking is a quantification.

Answer: We rewrote this section in the ms. We agree with the Reviewer that the effect of contamination can logically be derived from previous work on HGT, but we are not sure connecting HGT to contamination this way leads to crystal clear predictions. We therefore used a more cautious phrasing in the text.

2. Page 6, lines 198-199 - I am puzzled by the statement in the text that 33% of the conterminator hits (I assume that the authors here actually refer to the assignment as contamination) have “(...) taxon support values above 0.75 indicating possible false positives among those hits.” (line 199). How does this reconcile with Fig. 3a where the Venn diagram shows that all but 128 assignments are shared with ContScout. Is ContScout the also wrong? Note, the same is true for Basta. It might

be that I am overlooking something here, but this needs to be clarified. In essence, both alternative tools predict a subset of the ContScout contaminations. What is then the relevance of referring to 'possible false positives'?

Answer: The percentages of potential false positives are calculated for those proteins that were exclusively marked by one tool or the other (i.e. 33,196 hits reported only by ContScout {CS}, 343 hits reported only by BASTA {BA} and 117 hits reported exclusively by Conterminator {CT}.) Authors agree that wording on line 198 was misleading and amended the text in the ms.

Additionally, histograms of taxon support values for proteins exclusively detected by each tool are provided below for a quick comparison of potential false positive rates among tested tools. Percentage of tool-specific hits with 0% query taxon support (i.e. "confirmed" unique hits) are marked in green. Percentages of tool-specific hits with all reference matches confirming query taxon are marked in red (i.e. assumed false positive hits).

3. Along the same lines, how many proteins are flagged as contaminants without further evidence besides its placement together with proteins that were identified as contaminants using sequence similarity to database entries (impact of consensus call).

Answer: We added Supplementary Table 6 with the number of proteins that matched the expected query taxon at individual protein level yet were subsequently removed by ContScout because of the contig context. The added table contains data from the 200 most contaminated genomes, filtered at high level taxa (bacteria, animals, fungi, plants, other eukaryotes). The number of host-resembling proteins that were removed due to contig context remained low for all tested genomes.

*We further analysed this protein group (n=175) in *P. xuthus* where a manually filtered new draft genome version (GCF_000836235.2_Pxut_1.1) has been recently released. Although the best hits for these proteins suggested them to fit well in the query genome (i.e. tagged as metazoa), a throughout manual checking further high-scoring hits revealed that 71% of them also had high-scoring hits towards Microsporidia proteins, which turned out to be the single source of a massive contamination in *P. xuthus*. Moreover, all the host-resembling proteins that were removed by ContScout in *P. xuthus* were also manually removed during the genome version update.*

We believe that these examples clearly demonstrate the benefit of summarising taxon calls on the contig / scaffold level, as well as the robustness of ContScout.

4. Line 215 “Sensitivity assessment on manually filtered genomes”. The analyses attest ContScout an excellent sensitivity that exceeds that of Conterminator and the other tools by far. What is missing here, however, is the specificity assignment. This would help the readers a lot to gain confidence in this novel method.

*Answer: We agree that false positive rate should be mentioned in the ms. and added Table 1 to the main text with specificity data on all tested tools with two genomes, *Aspergillus zonatus* and *Papillus xuthus*.*

Minor issues

1. Page 2, line 64

a. please give a reference for a database-free taxonomic assignment tool

Answer: We agree with the Reviewer that “reference-free taxonomic assignment tool” sounds rather odd since a full-scale taxonomic assignment by definition needs some sort of reference data even though sorting sequences to individual bins based on k-mer analysis alone is possible. The review article of Cornet and Baurain

(doi:10.1186/s13059-022-02619-9) -which we consulted while writing the introduction section- lists ProDeGe, Anvivo and Blobtools as database-free k-mer based decontaminator tools. However, a close inspection of software descriptions (publications and documentations) reveal that all these tools in fact rely on reference databases at certain steps. Thus, the term “database-free” has been removed from the sentence in the ms.

b. Placing Blast searches and k-mers next to each other connected by an ‘or’ suggests that they are two alternative approaches, which they are not. Blast is the name of a database search heuristic--which in fact also makes use of k-mers to speed up the search for promising hit sequences—whereas pure k-mer-based approaches are just relying on exact pattern matching.

Answer: We agree with the Reviewer and rewrote this sentence.

2. Page 2, line 77 – I do not see the relevance of mentioning the phylogenomics studies here. The focus is on decontamination. What kind of downstream analysis a user has in mind, and whether this is done with a nucleotide- or amino acid alphabet, should be irrelevant here.

Answer: we removed the second part of the sentence from the ms.

3. Page 2, lines 80-82. From what was written before, it becomes not clear why the authors conclude that efficient and sensitive tools are ‘currently lacking’. The results of the present study should not serve as a justification for this statement in the introduction.

Answer: we rephrased this sentence.

4. Page 3, line 83 – I suggest removing the word ‘accurate’. This is a relative term and whether something is considered accurate depends on an ad hoc threshold.

Answer: Done.

5. Caption Figure 1

a. taxonomy should be rephrased to taxonomic assignment. A query genome has no ‘taxonomy’ (<https://en.wikipedia.org/wiki/Taxonomy>).

Answer: Caption for Figure 1 modified.

b. Case q in 1B is not ‘all green’ unless it is explained before what the meaning of the box is

Answer: Thank you for spotting the conflict between case1 figure and its explanation in the legend. Corresponding caption text has been modified.

c. The y axis of the bar charts should be explained

Answer: a brief description on bar plots that illustrate Case 1-4 has been added to Figure 1 Caption.

6. Page 4, lines 141-142 – I think that the term ‘biologically realistic contamination scenarios’ is not really helpful. In essence, contamination is a lab-issue and has little to do with biology.

Answer: We meant that these contamination scenarios were inspired by biology, such as pairs of frequently co-existing species which are easily ground up together during a DNA extraction (e.g. a parasite or commensal of a larger animal). Accordingly, we rephrased these occurrences to ‘biologically inspired’.

7. Page 4, Line 147 – The reference to Fig. 1 is wrong here

Answer: Corrected, thank you.

Minor discretionary Revisions

1. Fig. 2 – I am not happy with the term ‘host species’. It is tempting to interpret this in a biological way, but this would be inappropriate.

Answer: We aimed to eliminate the term ‘host species’ from the ms. However, since we could not come up with a better term for ‘host’ this is still used in the ms in some places.

Reviewer #2 (Remarks to the Author):

#1

I would like to express my gratitude to the authors for incorporating the modifications into the manuscript. The section on finer taxonomic affiliations is satisfactory. However, I still have serious reservations about this study, particularly regarding the distinction between HGT and contamination. The authors demonstrate that ContScout does not confuse HGT and contamination at very broad taxonomic levels, especially in bacteria. However, it is precisely in bacteria where HGT is most frequent, and this is even more true at lower taxonomic levels. Relying on the new analyses to claim that ContScout distinguishes HGT and contamination is insufficient.

Answer: We appreciate the Reviewer's suggestions. With regard to HGT we would like to point out that in the Discussion section of the ms (line 408-409 of the previous version) we used a cautious phrasing, emphasising that ContScout could distinguish HGT from contamination 'in the context of the analyzed empirical examples'. In the revised ms we provide further evidence that ContScout can distinguish HGT from contamination using ten known bacterial-bacterial HGT gene examples from the published work of Apjok et al, 2023 (doi: 10.1038/s41564-023-01320-2).

In nine out of ten cases, ContScout did not remove the known HGT gene at any tested taxon level (Supplementary Table 3). The only exception was the SAT4 streptothricin acetyltransferase ("CardRes_01521"), which was consistently removed by ContScout at family, order, class and phylum levels. A close inspection of the removed 3.8 kb contig identified a short repeated segment that we recognized as an indicator of a circular topology suggesting that the contig corresponds to a circular plasmid. SAT-4 is a known plasmid-mediated resistance determinant that can be present in the recipient bacteria either integrated in the genome or in the form of a circular plasmid.

When a HGT is present in the plasmid form, it is unlikely to contain any gene from the recipient genome thus it will be recognized as contamination by ContScout. Results and discussion section of the ms as well as the user manual of the tool has been modified to include this known limitation.

We note that, like our previous HGT analysis, and as we stress in the ms, these are examples that demonstrate the behaviour of ContScout, not a systematic analysis of HGT. The latter is beyond the scope of this paper.

#2

It was also clear in my review that, from my perspective, ContScout needed improvement in its algorithm to produce genomes as well, in order to meet the needs of the field. Eliminating contaminated proteins without addressing genomics is insufficient. Masking the corresponding part of the genome is an absolutely necessary task in my opinion for a publication.

*Answer: We modified the program code by adding a command line switch "-G" so that all contigs that are flagged as contamination can be removed both from the DNA sequence file and from the GTF / GFF file. That way, whenever the -G switch is enabled, six output files are generated at each taxon level in the output folder: a series of files that correspond to the filtered target genome (*_kept_dna.fasta, *_kept_annot.gff, *_kept_proteins.fasta) and a separate set that contains all data for the contamination (*_dropped_dna.fasta, *_dropped_annot.gff, *_dropped_proteins.fasta).*

I also requested the possibility for the authors not to eliminate the entire contig but only the relevant protein(s), as I am concerned about the loss of important data. I did not request this as the default mode but as an option for the user. The response to

this comment, "performed poorly in our benchmarks," is insufficient. I would have appreciated seeing these tests or even better, this option.

Answer: We refer the Reviewer to the answer to Remark#3 of Reviewer#1. There, we show that the number of proteins that match query taxon yet are removed on the 200-genome data set because of contig context is low.

*Furthermore, through the example of *Papilio xuthus*, where the ground truth is known, we found that all 175 host-resembling proteins removed by ContScout in fact correspond to proteins of truly foreign origin miscalled in the individual protein taxon labelling step.*

In our opinion, summarising protein calls over the contig and then handling all linked proteins together (including alien, ambiguous, unknown proteins as well as a few genuine-looking proteins) is one of the key improvements of ContScout that differentiates it from pre-existing tools and contributes to its demonstrated high sensitivity.

Nevertheless, there is an option (-n) that instructs the tool to ignore all annotation data and force filtering to take place entirely on the individual protein level. In order to demonstrate the effect of this operation mode, we analysed five protist genomes with or without using the -n option. In these tests, we focused on bacterial contamination by filtering at the superkingdom rank. As summarised in the table below, disabling the contig-wise taxon summarising feature of the program always resulted in a notable overestimation of contamination.

Due to the shortage of time available for the revision, we disagree to immediately implement the extra feature requested by the Referee that would allow masking / filtering contaminant proteins from alien contigs while keeping those contig parts / proteins that seem to match host taxon.

ShortName	Species	TaxonID	Normal operation	Annotation ignored	Fold overestimation	Comment
Phytmega	Phytophthora megakarya	4795	61	167	2.7	
Planfung	Planoprotostelium fungivorum	1890364	120	774	6.4	
Symbmicr	Symbiodinium microadriaticum	2951	115	134	1.2	
Thraclav	Thraustotheca clavata	74557	4	28	7	
Salprose	Salpingoeca rosetta	946362	11	430	39.1	Matriano et al report >175 HGT genes of bacterial origin for Salprose. DOI: 10.1038/s41598-021-85259-6

#3

Furthermore, it seems straightforward to infer a contamination rate based on the results of ContScout and compare it to EukCC. I understand that EukCC operates differently, but a comparison would have been interesting. Dismissing my comment based on the fact that EukCC is a QC tool is not a sufficient response for me.

Answer: As requested by the Reviewer, we performed EukCC analysis on eight genomes and compared the obtained contamination rates with values calculated based on ContScout outputs. Among the tested genomes, for Aspergillus zonatus, the ground truth is known. There, EukCC underestimates contamination by more than 50% (Truth: 12,99% EukCC: 6.02%, ContScout: 12,99%). For this genome, we carried out the analysis both in DNA (--DNA) and in Protein (--AA) mode but the results were identical. For the rest of the comparisons, we only performed EukCC contamination assessment runs on the protein (--AA) level, as explained below.

With Quercus suber, we were surprised to see that EukCC largely overestimated the extent of contamination (82% as compared to 25%, obtained on ContScout data). In line with that, in our hands, EukCC seemed to systematically overestimate the contamination for all species that had full genome duplications (primarily plants, Arachis hypogaea EukCC: 97.78% vs ContScout: 0%, Brassica napus, EukCC: 95.56% vs ContScout: 0%) or protein isoforms (Homo sapiens EukCC: 93.33%, ContScout:0%).

We noticed that EukCC contamination percentage estimations were usually based on a limited number of markers (10-263, mean: 69) which could also result in an inaccurate contamination estimate.

Finally, we were shocked to learn that EukCC seems to lack any marker set for clade metazoa as the user-selectable --clade options only contains "fungi", "protozoa" or "plants". In line with that, markers of protozoa were automatically selected for all tested animal genomes (Papilio xuthus, Drosophila melanogaster, Homo sapiens).

Even though results of pairwise comparison between EukCC and ContScout are available, we feel that they shall not be included in the ms. because they would render the text lengthy and complex while adding little if any extra information regarding the accuracy of ContScout.

Species	Clade	EukCC Clade	EukCC DNA/AA	Num Markers	EukCC %Contam	CS %Contam	Comment
Aspergillus zonatus	fungi	fungi	AA	25	6.02	12.99	True contam: 12.99%
Aspergillus zonatus	fungi	fungi	DNA	10	6.02	12.99	True contam: 12.99%
Quercus suber	plant	plant	AA	51	82.22	25.63	

Arachis hypogaea	plant	plant	AA	94	97.78	0	allotetraploid genome
Brassica napus	plant	plant	AA	68	95.56	0	allopolyploid genome
Papilio xuthus	animal	protozoa	AA	42	41.18	11.42	True contam: 11.70%
Drosophila melanogaster	animal	protozoa	AA	57	41.18	0	
Homo sapiens	animal	protozoa	AA	263	93.33	0	version with isoforms
Mucor circinelloides	fungi	fungi	AA	12	27.39	0	duplicated genome

#4

In summary, I do not recommend this paper for publication because my comments have not been adequately addressed. Modifying the algorithm to mask genomes, testing a non-greedy mode, comparing it to EukCC (which is used by many researchers working on fungi), and further analyzing HGT in bacteria vs bacteria are necessary tasks, in my opinion, for publication.

Answer: We would like to mention that the points the Reviewer listed here were novel requests and not problems with analyses already present in the ms. We evaluated each new request from the three Reviewers, with consideration of the hands-on time and analysis runtime required and did our best to decide which ones to implement, considering the limited time frame available for the revision. We hope the Reviewer understands that, as with any research project, additional new requests could be made endlessly with further interesting comparisons and analyses, but at some point the ms should be finalised and published.

In this revision round we implemented a switch (-G) in ContScout for the complete removal of contaminating contigs from the DNA sequence file as well as from the annotation in addition to the removal of all encoded proteins. At the same time, we respectfully disagree with the Reviewer's opinion to include the option of gene masking within alien-tagged contigs as our analysis shows such an option could negatively impact the analysis.

*As shown in Supplementary Table 6, added to the ms, proteins with true host origin that could be saved by this logic were generally rare, while in a specific example (*P. xuthus*), where the ground truth is known, all host-resembling proteins removed by ContScout turn out to belong to the contaminant (*Microsporidia*).*

Although the contig-level summarization feature of ContScout can be disabled via the switch "-n" (--no_annot), essentially yielding the "non-greedy" mode requested by the Reviewer, we strongly advise against its use as that would largely hit ContScout's performance. Firstly, ContScout with that option turned on removes

all HGT genes. It also results in a massive overestimation of contamination as shown in the table with five protist genomes above. Secondly, as stated in the user manual, the extent of agreement between proteins marked individually as alien and proteins removed after contig level summarization serves as an important quality control parameter that helps users to find the finest taxonomic resolution where contamination filtering can be meaningfully carried out.

Finally, I want to point out to the authors that for my second point, they partially referred me to the response to reviewer 1 without specifying which response. I assume it is referring to point 2. In this response, the authors refer to Figure XX, which is not very informative. In general, the response to the reviewers was difficult to follow.

Answer: We apologise for any inconvenience caused by the erroneous reference to Figure "XX".

I would like to conclude by extending my utmost respect to the authors and expressing my gratitude for their efforts. I want to emphasize that throughout this process, I have maintained a professional approach and strived to be meticulous.

Answer: We appreciate the Reviewer's criticism of our manuscript. We tried our best to address as many of the requests as possible in this and the previous revision round, however, we also note that there were considerably larger numbers and more diverse reviewer requests than what is possible to address in the time frames given for the revisions. Therefore, we had to prioritise among requests, and had to skip a few comparisons or the implementation of some features. In a few cases, we felt that the requested feature would have a negative impact on the analysis thus we opted against including them in our software tool. We hope in this revision round we could provide better justification for why certain feature requests could not be implemented.

Reviewer #3 (Remarks to the Author):

I would like to congratulate the authors for their thorough work on the resubmission and I feel the manuscript has greatly improved. Moreover, thank you for the updated code base and the improved user manual on the Github page. I am happy to report a successful installation and run of the algorithm.

Answer: We appreciate the Reviewer's comments and revised the ms in accordance with the suggestions.

However, I do have a few remaining concerns:

1) On line 374-377 the authors suggest the usage of ContScout as a routine screening method after genome assembly. However as pointed out by reviewer 2 as this a protein-based tool this can only happen after gene annotation, the very last (and often laborious) step before submission. I would therefore frame this more as a useful tool when embarking on large-scale comparative genomics projects, such as the ancestral gene content analysis in the paper. The benefit of using a protein methodology is the slower mutation rate of proteins vs DNA (line 73-75). However, to the best of my knowledge, I don't think the detection limits of nucleotide / protein divergence have been compared in the framework of contamination screening, especially relative to latest innovations in k-mer methods (such as FCS, which allows variation in the third-base position, see <https://www.biorxiv.org/content/10.1101/2023.06.02.543519v1.full>). The response of the authors on my second comment that these tools expect the genome of the contaminant to be present in the database, is wrong. While I theoretically acknowledge the benefit of proteins, it would be nice to see this compared as ContScout also lists the scaffolds to be removed (could be a supplementary figure).

Answer: Regarding the first part of this question, we better emphasise in the revised ms that ContScout works on annotated genomes. Nevertheless, we think ContScout should be used even in single-genome publications, not only in large-scale comparative genomics, because decontamination of even single genomes, if annotation is available, is important. This is underscored by the case of Quercus suber (<https://www.nature.com/articles/sdata201869>), in which the paper reported a single, highly contaminated genome.

Regarding the second part, we have carried out FCS-GX analysis of three genomes including A. zonatus and P. xuthus genomes where the list of contamination contigs (ground truth) is known. Contigs without any annotated protein were excluded to allow for a meaningful comparison between the two tools.

ShortName	Species	Lineage	FCS-gx only	CS only	Both	Comment
Papixuth	Papilio xuthus	animal	19 (T:19)	3 (T:2)	124 (T:124)	T: overlap with ground truth
Aspz01	Aspergillus zonatus	fungi	0	0	14 (T:14)	T: overlap with ground truth
Quersube	Quercus suber	plant	6	85	542	35/35 fungi specific contigs found by both tools

Both tools removed all bacterial contigs from A. zonatus and managed to tag most of the Microsporidia contamination in P. xuthus. Likewise, predictions for Q. suber were highly concordant between the two tools with ContScout yielding slightly more hits.

We conclude that both tested tools seem accurate with FCS operating exclusively at the DNA level and ContScout dealing with predicted proteins backed up by protein annotation data. Results of comparison between FCS and CountScout were added in the manuscript as Supplementary Table 2

2) The analysis of ancestral gene content is heavily dependent on the topology. I didn't notice this last time, but the sister relationship between Amoebozoa and Discoba is unexpected (e.g. Burki 2020). Could the phylogeny be adapted similar as done in R1 comment 6?

Answer: We appreciate this comment, the topological inconsistency has escaped our attention too. We applied a constraint to the tree and recomputed gene duplication/loss statistics. See Figure 6 in the revised ms.

3) On line 437-438 you acknowledge that not only contamination but also incorrect gene annotation and HGT can impact ancestral gene reconstruction analyses. Next, you state that the latter two processes will mainly affect terminal branches and therefore have only a limited effect on gene losses. However I don't believe HGT, especially cross-kingdom HGT, mainly affect the distal portion of the tree. In a parsimonious method, one gain and several losses will be preferred over two independent gains. Could you please explain how does the effect of HGT compare to contamination on gene loss inflation levels?

Answer: We agree the second part of our sentence was only true for incomplete annotation, but not HGT. We rephrased this section.

Minor / Textual comments:

Line 43: add comma after fungi

Answer: Done.

Line 49: 'while sample mishandling and incorrect data processing can cause technical problems'. You have listed here a set of problems that can result in contamination. Sentence doesn't have a logical causal relationship.

Answer: section removed from the text.

Line 73: Please add closing parenthesis

Answer: Done.

Line 100: add the word 'or'

Answer: Done.

Line 252: scrutinizing instead of scrutinize. Please split this sentence in two to help readability.

Answer: Done.

Line 303: Please add closing parenthesis

Answer: Done.

On line 226 you mention 680 contaminant proteins in the *B. impatiens* genome, while on line 316 you mention 965. Is due to the existence of two different contaminated genome versions?

*Answer: Both numbers are accurate but they describe different protein sets. Number **965** refers to all bacterial proteins identified by ContScout in Bombimpa. Number **680**, on the other hand, refers to the manually curated bacterial proteins in the Bombimpa assembly, which we used as ground truth. We believe we highlighted this difference in the ms.*

Line 345: add space between as and bacterial.

Answer: Done.

Line 439: Please split the sentence after the reference.

Answer: Done.

Section sensitivity assessment on manually filtered genomes:

Could you please mention how many (if any) genes were flagged by any of the tools that weren't manually removed in the datasets.

*Answer: We added specificity data for all tools on the *A. zonatus* (Aspzo1) genome in the ms as Table1. Additionally, we included a new genomic set for *P. xuthus* (Papixuth) that was used as an additional gold standard. All hits marked by Conterminator, BASTA, MMSeqs and Diamond were true positives. Among the hits of ContScout, there was one false positive from the Papixuth genome and zero from Aspzo1.*

REVIEWERS' COMMENTS

Reviewer #1 (Remarks to the Author):

I have no further comments

Reviewer #2 (Remarks to the Author):

I thank the authors for their efforts on the manuscript and the algorithm. Like the authors, I am surprised by the performance of EukCC; this could have been a subject for a support note, but I leave it to the authors' discretion as I know that negative comments on other algorithms are sometimes poorly received. I am convinced by the responses provided by the authors in this review, and I recommend the publication of this work. ContScout is a very interesting tool, congratulations.

Reviewer #3 (Remarks to the Author):

In the latest revised version of this manuscript the authors have made substantial changes to the manuscript. I especially appreciated the novel software option that was suggested by R2 and the clarification of the text throughout the manuscript. I support the publication of the manuscript in its current state and would only suggest removing 'As per reviewer request' on line 242 page 7.

RESPONSE TO REVIEWERS' COMMENTS

The authors would like to thank the three reviewers for the time and effort they invested in the review process. Their feedback and recommendations have reshaped and greatly improved both the manuscript and the programme code. They have inspired us to add new features and modifications that largely increased the usability and usefulness of ContScout.